# The rostroventral part of the thalamic reticular nucleus modulates fear extinction

Joon-Hyuk Lee[1], Charles-Francois V. Latchoumane[1], Jungjoon Park[1,2], Jinhyun Kim[3], Jaeseung Jeong[2], Kwang-Hyung Lee[2] & Hee-Sup Shin[1]*

The thalamus has been implicated in fear extinction, yet the role of the thalamic reticular nucleus (TRN) in this process remains unclear. Here, in mice, we show that the rostroventral part of the TRN (TRNrv) is critically involved in the extinction of tone-dependent fear memory. Optogenetic excitation of TRNrv neurons during extinction learning dramatically facilitated, whereas the inhibition disrupted, the fear extinction. Single unit recordings demonstrated that TRNrv neurons selectively respond to conditioned stimuli but not to neutral stimuli. TRNrv neurons suppressed the spiking activity of the medial part of the dorsal midline thalamus (dMTm), and a blockade of this inhibitory pathway disrupted fear extinction. Finally, we found that the suppression of dMTm projections to the central amygdala promotes fear extinction, and TRNrv neurons have direct connections to this pathway. Our results uncover a previously unknown function of the TRN and delineate the neural circuit for thalamic control of fear memory.

[1] Center for Cognition and Sociality, Institute for Basic Science (IBS), Daejeon 305-338, Korea. [2] Department of Bio and Brain Engineering, Korea Advanced Institute of Science and Technology (KAIST), Daejeon 305-338, Korea. [3] Center for Functional Connectomics, Korea Institute of Science and Technology, Seoul 136-791, Korea. *email: shin@ibs.re.kr

Recent thalamic studies have revealed higher cognitive functions of the thalamus beyond sensory relay[1,2], one of which is the control of fear memory. Thus recent studies revealed a critical role of the limbic thalamus in persistent attenuation of fear by using pharmacological and optogenetic manipulations[3] and in fear extinction by using genetic[4] and chemogenetic[5] manipulations. The thalamic reticular nucleus (TRN), a shell of GABAergic neurons surrounding the thalamus, provides monosynaptic inhibitory inputs to the thalamus[6,7], thus capable of suppressing inappropriate thalamic signals by inhibiting the thalamus in a timely manner[8–12]. Despite the importance of the TRN in thalamic information processing[13], the role of the TRN in control of fear memory has not been explored. Previous studies have demonstrated that the TRN consists of several sectors including sensory (gustatory, somatosensory, visceral, visual, and auditory) and limbic sectors[14]. An anatomical study in primates showed that the limbic sector receives input from the amygdala[15], suggesting a potential role of the TRN in the control of fear. Nevertheless, there is no experimental evidence to support this notion.

Recent studies showed that the dorsal midline thalamus (dMT), which includes the paraventricular nucleus of the thalamus (PVT) and the mediodorsal thalamic nucleus (MD)[3,16], plays a critical role in persistent attenuation of fear[3], whereas it is not involved in fear conditioning[16]. These studies suggest a possibility that a specific TRN area corresponding to the dMT might play a crucial role in fear extinction, yet, this possibility has not been investigated.

In current study, as the first step to explore the role of the TRN in fear extinction, we identified that the rostroventral part of the TRN (TRNrv) distinctly projects to the medial part of the dMT (dMTm) which was previously implicated in persistent attenuation of fear[3]. Guided by this connection map, we carried out circuit analysis and found that TRNrv neurons are activated during extinction learning. The TRNrv neurons suppressed the spiking activity of the dMTm neurons, and this suppression was required for fear extinction. Furthermore, we demonstrate that the suppression of the dMTm projections to the central amygdala (CeA) promotes fear extinction and that TRNrv neurons have direct synaptic connections to this dMTm–CeA circuit. These results show the critical role of the TRN in fear extinction and reveal a novel neural pathway underlying fear extinction.

## Results

**Distinct projection pattern of TRNrv neurons to the dMTm.** To examine the anatomical projection pattern of limbic sector of the TRN to the thalamus, we injected an adeno-associated virus (AAV), AAV9-DIO-GFP, into the rostral part of the TRN, where the limbic sector is located[7], in parvalbumin-Cre (PV-Cre) mice (Fig. 1a, b). Because the majority of TRN neurons are PV-positive[17,18], we could infect the majority of TRN neurons by targeting PV neurons using cre-dependent system. Also, we could specifically infect only TRN neurons without infecting nearby neurons due to the absence of PV-positive neurons near the TRN[19]. For precise validation of the constrained expression of the virus within the TRN, we used PV immunostaining[19] (Fig. 1b). After virus injection into the rostral part of the TRN, we observed green fluorescent protein (GFP)-labeled axons throughout the dMT (Fig. 1d). To define finer connectivity between the rostral part of the TRN and the dMT, we simultaneously injected two different retrograde tracers: fluorogold (FG) into the medial part of the dMT (dMTm), which covers the PVT and the medial part of the MD (Fig. 1f), and cholera toxin-B subunit (CTB) into the lateral part of the dMT (dMTl), which covers the lateral part of the MD and the centrolateral thalamic nucleus (CL) (Fig. 1e–i).

The injection positions were confirmed by calbindin-D28k (CB) immunostaining (Fig. 1f) which is known to delineate the boundaries of limbic structures[19]. As a result, we observed the FG signals in TRNrv neurons (Fig. 1k), whereas CTB signals were found in rostrodorsal part of the TRN (TRNrd) neurons (Fig. 1l). We also observed consistent results when the injection positions of the two tracers were switched (Supplementary Fig. 1). Together, these results indicate that TRNrv neurons project to the dMTm, whereas TRNrd neurons project to the dMTl, in a mutually exclusive way.

**Manipulations of TRNrv neurons affect fear extinction.** Next, we examined whether TRNrv neurons or TRNrd neurons are involved in fear extinction by optogenetic manipulation of PV neurons in either the TRNrv or the TRNrd during fear extinction learning (Fig. 2a, b). Since the TRNrd and TRNrv are closely located, precise spatial targeting of the light stimulation was important to achieve the goal. To do this, we delivered the light at the minimum intensity (see "Methods" section: Optogenetic stimulation) and with precise positioning of the optical fiber (Fig. 2c, e, g). For the control groups in all optogenetic experiments, we injected the same virus as for the experimental group and blocked the light transmission into the brain (Supplementary Fig. 3, see "Methods" section: Optogenetic stimulation). Our results showed that optogenetic excitation of TRNrv PV neurons using a channelrhodopsin-2 (ChR2) virus during extinction learning significantly reduced freezing level during extinction learning (Fig. 2d, Day 2, 2nd–18th tones of Extinct., two-way repeated-measures analysis of variance (RM ANOVA) followed by Holm–Sidak method, $F_{(1, 23)} = 21.555$, $P < 0.001$, see Supplementary Movie 1). The difference between the control group and the stimulated group was not due to different levels of fear memory acquisition because both groups showed similar levels of freezing to the first tone in extinction learning in which both groups were free from the optogenetic stimulation (Fig. 2d, first tone on Day 2, two-tailed $t$ test, $t_{(23)} = -0.147$). Notably, this reduced freezing level was persistently observed in the retrieval test on Day 3 (Fig. 2d, Day 3, four tones of the test day, two-way RM ANOVA followed by Holm–Sidak method, $F_{(1, 23)} = 54.679$, $P < 0.001$, see Supplementary Movie 1), during which no light stimulation was delivered. Consistently, inhibition of the TRNrv PV neurons using the AAV9-DIO-Arch-GFP virus induced an elevated freezing level during both the extinction learning on Day 2 and the retrieval test on Day 3 (Fig. 2f, Day 2, 2nd–18th tones of Extinct., two-way RM ANOVA followed by Holm–Sidak method, $F_{(1, 25)} = 6.055$, $P = 0.021$, Day 3, four tones of the test day, two-way RM ANOVA followed by Holm–Sidak method, $F_{(1, 25)} = 10.442$, $P = 0.003$). However, inhibition of the TRNrd PV neurons had no effect on freezing behavior during either the extinction learning on Day 2 or the retrieval test on Day 3 (Fig. 2h, Day 2, 2nd–18th tones of Extinct., two-way RM ANOVA, $F_{(1, 15)} = 0.0737$, $P = 0.790$, Day 3, four tones of the test day, two-way RM ANOVA, $F_{(1, 15)} = 0.0369$, $P = 0.850$). We further tested whether the TRNrv is involved in fear acquisition by excitation of the TRNrv during fear conditioning, and we did not observe significant changes in freezing responses during either the conditioning on Day 1 or fear retrieval test on Day 2 (Supplementary Fig. 4, Day 1, Condi., two-way RM ANOVA, $F_{(1, 12)} = 0.0387$, $P = 0.847$, Day 2, Test, two-tailed $t$ test, $t_{(12)} = 0.902$). Together, these results suggest that optogenetic excitation of TRNrv PV neurons is sufficient to enhance fear extinction, and the activity of PV neurons in the TRNrv, but not in the TRNrd, is required for fear extinction.

It is noteworthy that we applied a stronger shock (0.5 mA) during the fear conditioning in the excitation experiment (Fig. 2d) to avoid floor effect. This induced a higher freezing level

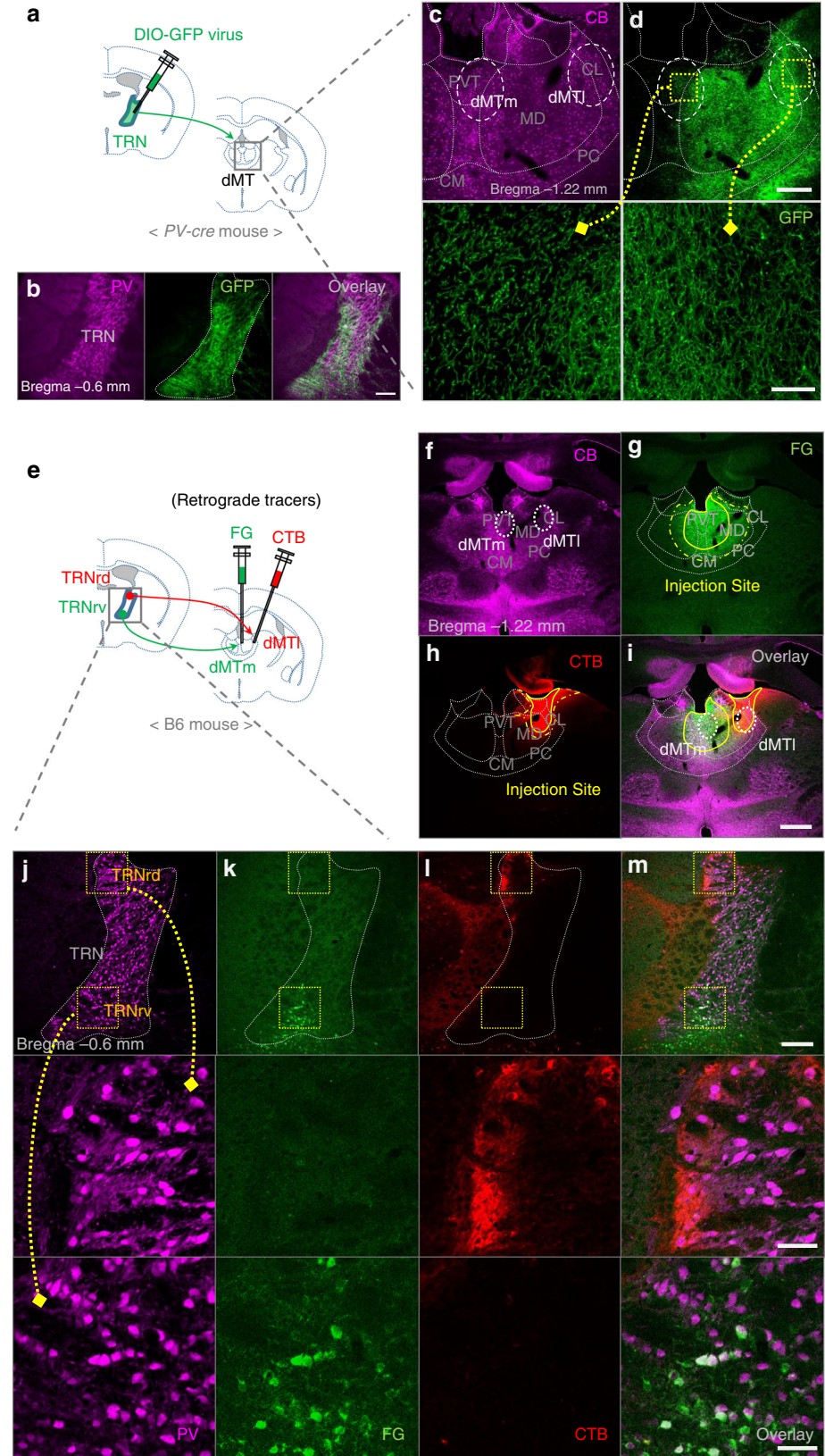

(Supplementary Fig. 2), thus allowing clear detection of freezing suppression. For similar reason, we applied a weaker shock (0.3 mA) in the inhibition experiments (Fig. 2f, h) to avoid ceiling effect, as the previous study employed[20]. As an exploration, we also tested 1 Hz optogenetic excitation; however, compared to the effect of 10 Hz optogenetic stimulation (Fig. 2d), 1 Hz optogenetic excitation of the TRNrv PV neurons did not affect fear extinction (Supplementary Fig. 5a, b). Both 1 and 10 Hz optogenetic stimulation of the TRNrv PV neurons did not affect either locomotor activity (Supplementary Fig. 5c, two-way RM ANOVA, $F_{(2, 31)} = 0.118$, $P = 0.889$) or anxiety level (Supplementary Fig. 5d, one-way ANOVA, $F_{(2)} = 1.816$, $P = 0.18$).

**Fig. 1** Distinct projection patterns of the rostroventral part of the TRN (TRNrv) and the rostrodorsal part of the TRN (TRNrd). **a** Schematic depiction of virus injection. **b** The neurons in the rostral part of the TRN are infected by DIO-GFP virus. Left, PV immunostaining delineates the boundary of the TRN. Scale bar, 200 μm. **c** CB immunostaining delineates the boundary of the dorsal midline thalamus (dMT). The medial part of the dMT (dMTm) and the lateral part of the dMT (dMTl) are marked with dotted white circles. **d** GFP signals of axonal projections from the rostral part of the TRN are observed across the dMTm and the dMTl. Scale bar, 200 μm. Magnified images of yellow rectangles are shown in the lower panels. Scale bar, 50 μm. **e** Schematic depiction of injections of retrograde tracers. **f** CB immunostaining delineates the boundary of the dMT. **g, h** injection sites of FG (**g**) and CTB (**h**). Solid and dotted yellow lines indicate the sites where the strong and the weak signals were observed, respectively. **i** Overlaid image. Scale bar, 500 μm. **j** PV immunostaining delineates the boundary of the TRN. Magnified images are shown in lower panels. **k** FG signals are observed only in the TRNrv but not in the TRNrd. **l** CTB signals are observed only in the TRNrd but not in the TRNrv. **m** Overlaid image. Scale bar, 200 μm. Magnified images. Scale bar, 50 μm. PVT paraventricular nucleus of the thalamus, MD mediodorsal thalamic nucleus, CL centrolateral nucleus, PC paracentral thalamic nucleus, CM central medial thalamic nucleus

**Selective response of the TRNrv to fear-conditioned stimulus.** Because fear extinction was affected by the optogenetic manipulations that may have changed the firing activities in the TRNrv, we next examined the firing activity of TRNrv neurons during fear extinction learning in vivo. We injected DIO-ChR2 virus and implanted an optrode into the TRNrv of *PV-cre* mice. To identify whether the recorded TRNrv neurons were PV-positive, we performed tagging procedure 5 min before the first tone presentation of extinction learning (Fig. 3a). In the tagging procedure, we applied 10 Hz light stimulation to TRNrv neurons for 30 s. If a neuron showed increased spiking activity to the light stimulation (Fig. 3b), which indicated that this neuron expresses ChR2 and is PV-positive, we named it as a photo-tagged PV (PP) neuron. The proportion of PP neurons among the recorded TRNrv neurons was 32% (Fig. 3f, left panel).

We also recorded spiking responses of TRNrv neurons to the neutral tones, i.e., the tones before they were associated with the fear, and compared them to the spiking responses to the conditioned tones during extinction learning (Fig. 3c). The reason was that we did not exclude the possibility that the TRNrv might simply respond to sensory stimulus itself, which is auditory tone, whether or not the sensory stimulus is associated with the fear. We also applied the tagging protocol before the first presentation of neutral tone, and the proportion of PP neurons among the recorded TRNrv neurons was 24% (Fig. 3d, left panel).

As a result, notably, TRNrv neurons showed increased firing activity selectively to the conditioned tones (Fig. 3f, first row, ALL neurons) but not to the neutral tones (Fig. 3d, first row, ALL neurons). We compared the responses of the baseline (5 s before the tone) and 5 s following the start of the tone, and there was significant increase in firing rate only by the conditioned tones (Fig. 3g, top panel, one-sample signed-rank test, $Z = 6.074$, $P < 0.001$) but not by the neutral tones (Fig. 3e, top panel, one-sample signed-rank test, $Z = 1.432$, $P = 0.154$). This selective response was also observed in the analysis of PP population (Fig. 3d, blue color in the bottom panel, two-tailed one-sample $t$ test, $t_{(9)} = 0.105$, $P = 0.919$; Fig. 3g, blue color in bottom panel, one-sample signed-rank test, $Z = 3.667$, $P < 0.001$).

To manifest the actual firing change caused by our optogenetic excitation during extinction learning, which is related to facilitation of fear extinction (Fig. 2d), we examined how the spiking responses of TRNrv neurons are changed by our 10 Hz optogenetic stimulation during extinction learning (Supplementary Fig. 6c, d, bright colors). In the analysis of PP population, we observed significant increase of spiking responses in the stimulated group compared to the control group (Supplementary Fig. 6d, bright blue color for the stimulated group and dark blue color for the control group, Mann–Whitney rank-sum test, $U = 52.000$, $P = 0.001$).

There was a possibility that TRNrv neurons might not selectively respond to the conditioned tone but simply become sensitive to the neutral tone due to the repeated exposure of same tone after certain time. To test this possibility, we tested another batch of mice that were exposed to the neutral tones two times with the interval of 3 days (Supplementary Fig. 7a). The interval was equivalent to the interval between neutral tone and the conditioned tone (Fig. 3c). In addition, we also recorded spiking responses of TRNrd neurons to examine whether TRNrd neurons show any significant responses to conditioned tones. As a result, TRNrv neurons did not show significant increase of spiking activities to re-exposure of neutral tones (Supplementary Fig. 7c, d, black colors, TRNrv-Trial 1, two-tailed one-sample $t$ test, $t = 1.242$, $P = 0.243$; TRNrv-Trial 2, two-tailed one-sample $t$ test, $t = 0.697$, $P = 0.503$), and we again observed the increased firing activities of TRNrv neurons selectively by the conditioned tones (Supplementary Fig. 7e, black colors, one-sample signed-rank test, $Z = 3.133$, $P = 0.002$) as we previously observed (Fig. 3d, f). We did not observe significant changes of spiking activities of TRNrd neurons by either the neutral tones (Supplementary Fig. 7c, d, gray colors, TRNrd-Trial 1, two-tailed one-sample $t$ test, $t = 0.261$, $P = 0.811$; TRNrd-Trial 2, two-tailed one-sample $t$ test, $t = 1.169$, $P = 0.281$) or the conditioned tones (Supplementary Fig. 7e, gray colors, two-tailed one-sample $t$ test, $t = 2.312$, $P = 0.0687$), which is consistent with our behavioral result showing that the inhibition of TRNrd neurons during extinction learning does not affect fear extinction (Fig. 2h).

Together, these results show that TRNrv neurons are selectively activated by fear-related cue but not by neutral cue, whereas TRNrd neurons are not responsive to either of the cues.

**The suppression of dMTm firing activity by the TRNrv.** To test the physiological effect of TRNrv neurons to dMTm neurons in vivo, we recorded spiking responses of dMTm neurons while we optogenetically excited TRNrv neurons (Fig. 4a). We observed that 78.1% of dMTm neurons were robustly inhibited by 10 Hz optogenetic stimulation of TRNrv neurons (Fig. 4a, top right panel), while 21.9% neurons were not affected (Fig. 4a, bottom right panel). We did not observe any dMTm neurons that were excited.

To examine whether TRNrv→dMTm inhibitory pathway is required in fear extinction, we optogenetically blocked this pathway during extinction learning (Fig. 4b). As a result, we observed elevated freezing response on retrieval test on Day 3 (Fig. 4c, Day 3, four tones of the test day, two-way RM ANOVA followed by Holm–Sidak method, $F_{(1, 25)} = 6.550$, $P = 0.017$), although, interestingly, we did not observe difference in freezing level during extinction learning on Day 2 (Fig. 4c, Day 2, 2nd–18th tones of Extinct., two-way RM ANOVA, $F_{(1, 25)} = 0.334$, $P = 0.568$). Together, these results indicate that the inhibitory projections of the TRNrv to the dMTm, which suppress firing activity of dMTm neurons, are required for fear extinction.

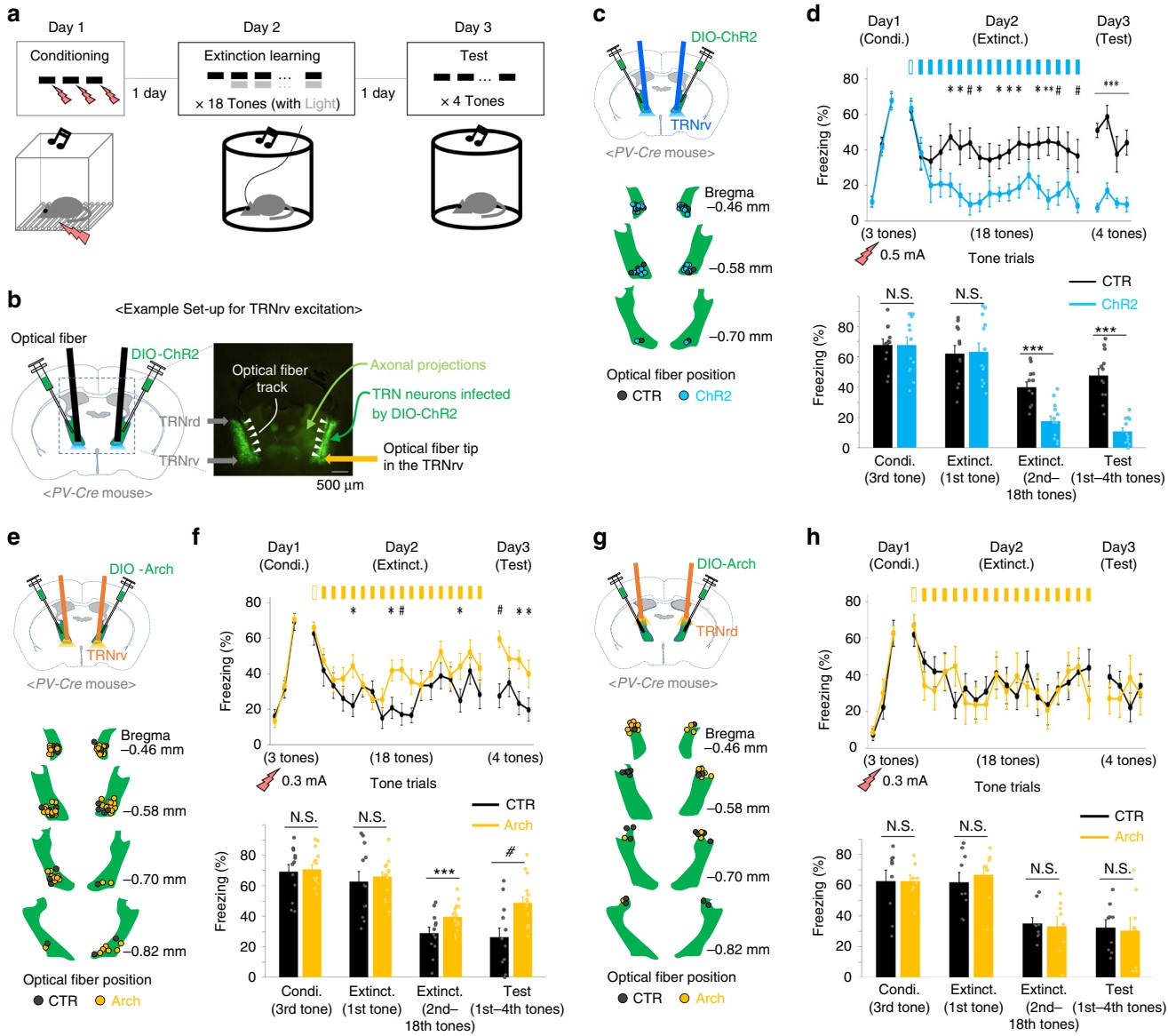

**Fig. 2** Behavioral results of optogenetic modulation of the TRNrv and the TRNrd during fear extinction learning. **a** Schematized behavioral design. **b** Schematic depiction and representative image for virus injection and fiber implantation are shown. **c** Schematic depiction and fiber positions are shown. **d** Top panel, Optogenetic excitation (450 nm light, 6.3 ms pulse duration, 10 Hz) of the TRNrv ($n = 12$ for control, $n = 13$ for stimulated group) induced decreased freezing levels during extinction learning (2nd–18th tones of Extinct., two-way RM ANOVA followed by Holm–Sidak method, $F_{(1, 23)} = 21.555$, $P < 0.001$) and during the retrieval test (four tones of the test day, two-way RM ANOVA followed by Holm–Sidak method, $F_{(1, 23)} = 54.679$, $P < 0.001$). Bottom panel, Quantified data of the top panel. Condi. conditioning, Extinct. extinction. **e** Schematic depiction and fiber positions are shown. **f** Top panel, Optogenetic inhibition (561 nm light, continuous pulse during the tone) of the TRNrv ($n = 12$ for control, $n = 15$ for the stimulated group) induced increased freezing levels during extinction learning (2nd–18th tones of Extinct., two-way RM ANOVA followed by Holm–Sidak method, $F_{(1, 25)} = 6.055$, $P = 0.021$) and during the retrieval test (four tones of the test day, two-way RM ANOVA followed by Holm–Sidak method, $F_{(1, 25)} = 10.442$, $P = 0.003$). Bottom panel, Quantified data of the top panel. **g** Schematic depiction and fiber positions are shown. **h** Top panel, Optogenetic inhibition (561 nm light, continuous pulse during the tone) of the TRNrd ($n = 9$ for control, $n = 8$ for the stimulated group) did not affect freezing levels during extinction learning (2nd–18th tones of Extinct., two-way RM ANOVA, $F_{(1, 15)} = 0.0737$, $P = 0.790$) and during the retrieval test (four tones of the test day, two-way RM ANOVA, $F_{(1, 15)} = 0.0369$, $P = 0.850$). Bottom panel, Quantified data of the top panel. All data are presented as mean ± SEM. N.S., not significant. *$P < 0.05$, #$P < 0.01$, **$P < 0.005$, ***$P < 0.001$. See Supplementary Table 1 for values of post hoc test

**The role of dMTm projections to the CeA in fear extinction**. It has been shown that the optogenetic inhibition of the PVT→CeA pathway during fear memory retrieval at 7 days after the conditioning induces persistent attenuation of fear[3]. Considering that the PVT is a part of the dMTm, this previous study raised a possibility that the TRNrv might exert its effect through the dMTm→CeA pathway. We expected that, if this is the case, the dMTm→CeA pathway would be involved in fear extinction. To

address this issue, we inhibited the dMTm→CeA pathway during extinction learning (Fig. 5a) after we injected CamKIIa-NpHR3.0 virus into the dMTm and implanted optical fibers into the CeA of B6 mice (Fig. 5b). In this experiment, we added one more control group in which we injected CamKIIa-EYFP virus and delivered the light during extinction learning (see "Methods" section: Optogenetic stimulation) to double confirm the behavioral change of the inhibited group. As a result, in retrieval test, the

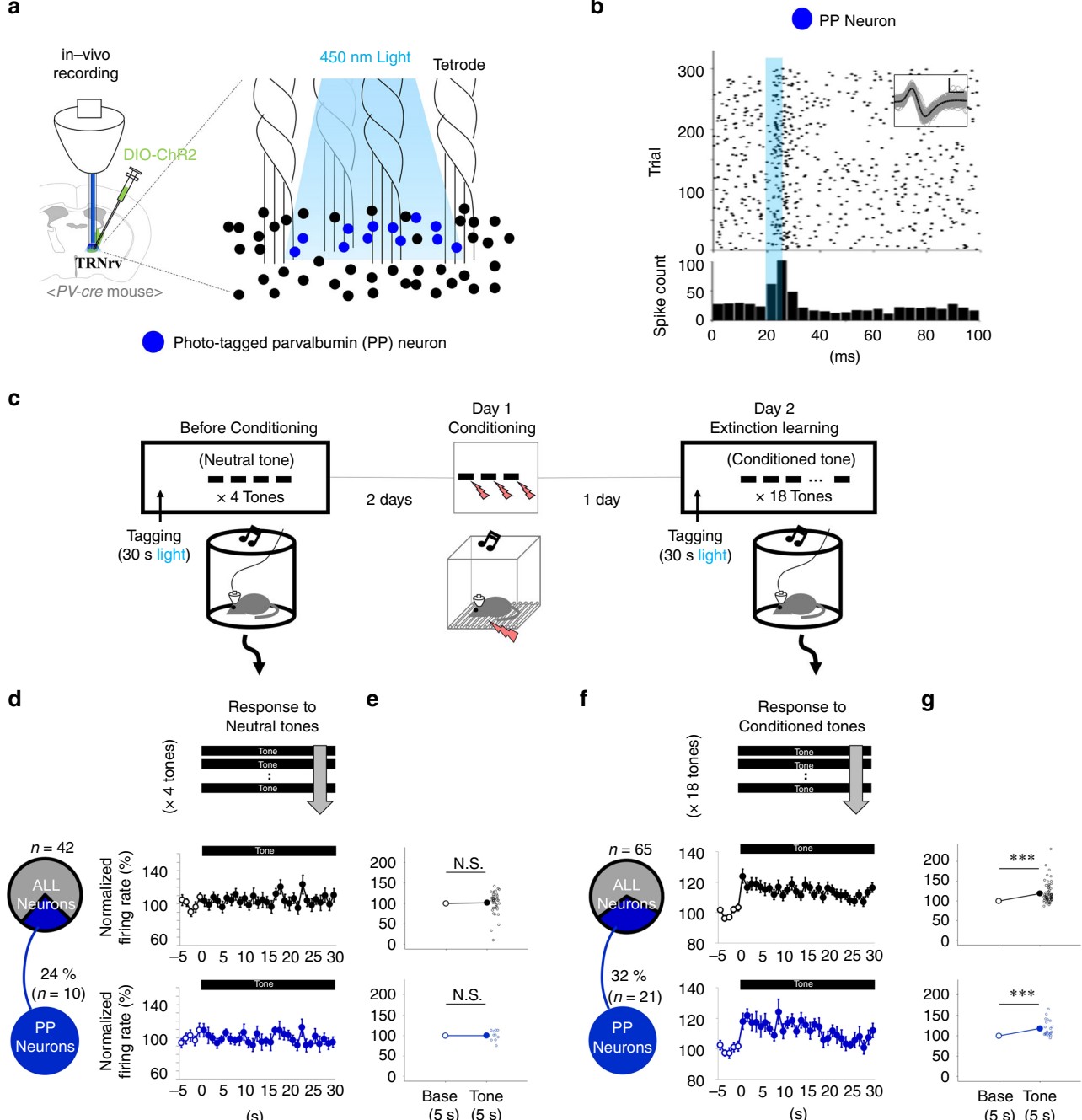

**Fig. 3** Spiking responses of TRNrv neurons to neutral stimulus and conditioned stimulus. **a** Schematic depiction of Optrode recording. The neurons responsive to the light stimulation were named as photo-tagged parvalbumin (PP) neurons. **b** Representative PP neuron shows time-locked spikes to light stimulations. Insets, spike shape of example neuron. Black lines for mean value, gray lines for individual spikes. Scale bar, 50 μV, 200 μs. **c** Two days before the conditioning, the spiking responses of TRNrv neurons to neutral tones, i.e., the tones not associated with the fear, were recorded. One day after the conditioning, the spiking responses of TRNrv neurons to conditioned tones, i.e., the tones associated with the fear, were recorded. For tagging procedure, brief light stimulations were delivered 5 min before the first tone of neutral tones or conditioned tones. **d** Normalized firing responses of TRNrv neurons to neutral tones. Data are shown for ALL and PP populations in each rows (ALL, $n = 8$ mice, 42 neurons; PP, $n = 5$ mice, 10 neurons). Right, responses of 5 s baseline and 30 s tone are shown. **e** Responses of 5 s baseline and first 5 s of the tone are compared. No significant differences were found (ALL neurons, one-sample signed rank test, $Z = 1.432$, $P = 0.154$; PP neurons, two-tailed one-sample $t$ test, $t = 0.105$, $P = 0.919$). **f** Normalized firing responses of the neurons to conditioned tones. Data are shown for ALL and PP populations in each rows (ALL, $n = 10$ mice, 65 neurons; PP, $n = 5$ mice, 21 neurons). Right, responses of 5 s baseline and 30 s tone are shown. **g** Responses of 5 s baseline and first 5 s of the tone are compared. Significant differences were observed in ALL and PP populations (ALL neurons, one-sample signed-rank test, $Z = 6.074$, $P < 0.001$; PP neurons, one-sample signed-rank test, $Z = 3.667$, $P < 0.001$). All data are presented as mean ± SEM. N.S., not significant. ***$P < 0.001$

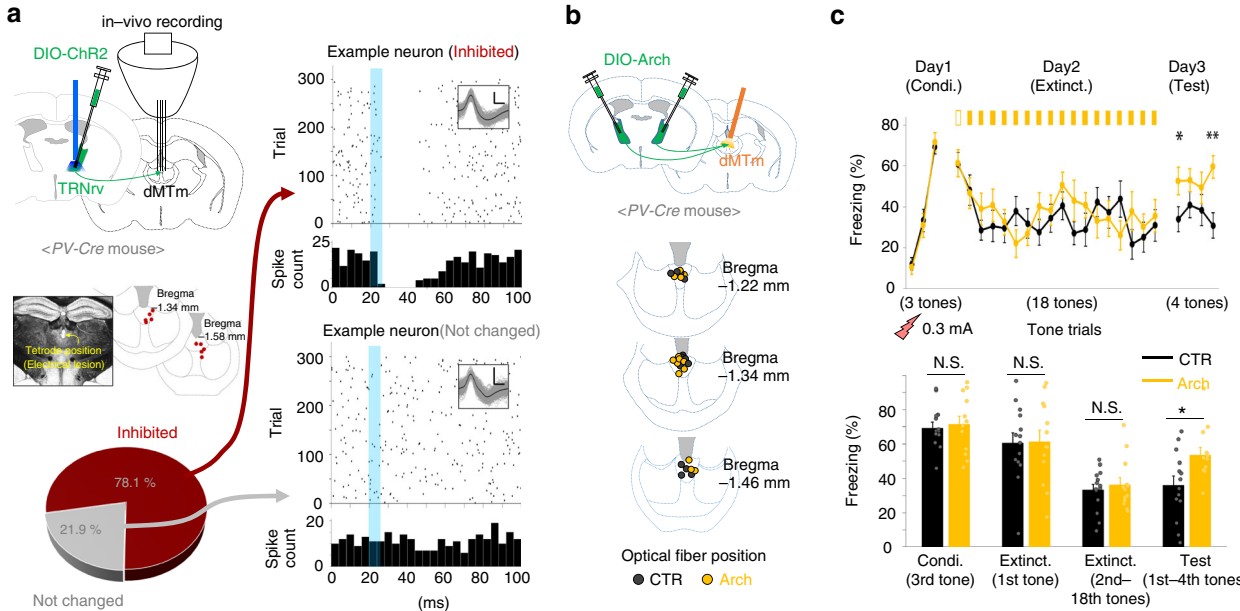

**Fig. 4** The suppression of spiking activity of dMTm neurons by TRNrv inhibitory inputs is required for fear extinction. **a** Left top, TRNrv neurons were infected by DIO-ChR2 virus. Optical fiber and tetrodes were implanted in the TRNrv and in the dMTm, respectively. Left middle, The electrode positions are shown. Left bottom, Different responses of dMTm neurons to TRNrv optogenetic excitation were observed (inhibited neurons, $n = 25$, unchanged neurons, $n = 7$). Right, Example neurons that show inhibition (upper panel) or unchanged response (lower panel) by optogenetic excitation of the TRNrv. Insets, spike shape of each example neurons. Black lines for mean value, gray lines for individual spikes. Scale bar, 50 μV, 200 μs. **b** Top, Schematic depiction for optogenetic inhibition of the TRNrv→dMTm pathway during fear extinction learning. Bottom, The positions of fiber tips are marked by yellow dots for the stimulated group and gray dots for the control group. **c** Top panel, Optogenetic inhibition (561 nm light, continuous pulse during the tone) of the TRNrv→dMTm pathway ($n = 14$ mice for the control group, $n = 13$ mice for the stimulated group) induced increased freezing levels during the retrieval test (four tones of the test day, two-way RM ANOVA followed by Holm–Sidak method, $F_{(1, 25)} = 6.550$, $P = 0.017$) but did not change freezing levels during extinction learning (2nd–18th tones of Extinct., two-way RM ANOVA, $F_{(1, 25)} = 0.334$, $P = 0.568$). Bottom panel, Quantified data of the top panel. All data are presented as mean ± SEM. N.S., not significant. *$P < 0.05$, **$P < 0.005$. See Supplementary Table 1 for values of post hoc test

inhibited group showed significantly reduced freezing level compared to either of the control groups (Fig. 5c, Day 3, four tones of the test day, two-way RM ANOVA followed by Holm–Sidak method, $F_{(2, 31)} = 5.214$, $P = 0.011$; Unstim. vs. NpHR, $t = 2.860$, $P = 0.022$; EYFP vs. NpHR, $t = 2.744$, $P = 0.02$; Unstim. vs. EYFP, $t = 0.0655$, $P = 0.948$), indicating that dMTm→CeA pathway is involved in fear extinction.

**Anatomical relation between the TRNrv, dMTm, and CeA.** Although we showed that both the TRNrv→dMTm pathway and the dMTm→CeA pathway affect fear extinction, we were not sure whether those two pathways overlap at the dMTm. By using retrograde virus (rAAV2-Retro-cre) and mono-trans-synaptic rabies virus (hSyn-FLEX-TVA-eGFP-oG, EnvA G-Deleted Rabies-mCherry) (Fig. 6a), we found that TRNrv neurons are disynaptically connected to the CeA through the dMTm (Fig. 6i–k). The restricted injection of retrograde virus into the CeA was confirmed by restricted axonal signals in the CeA (Fig. 6f–h) and the restricted expression of rabies virus in the dMTm was confirmed by GFP signals from helper virus (Fig. 6c) and mCherry signals from rabies virus in the dMTm (Fig. 6d). This result was successfully replicated in other mice (Supplementary Fig. 8). Together, these results indicate that TRNrv neurons are able to directly control dMTm→CeA circuit.

## Discussion

Here we show that the TRNrv modulates fear extinction. TRNrv neurons showed increased firing activities to the conditioned tones during fear extinction learning and optogenetic excitation

or inhibition of TRNrv neurons during extinction learning induced facilitation or disruption of fear extinction, respectively. The TRNrv neurons suppressed the spiking activities of dMTm neurons, and blockade of this suppression impaired fear extinction. Retrograde labeling study revealed that TRNrv neurons have direct synaptic connections to the dMTm→CeA circuit, and the suppression of this dMTm→CeA circuit promoted fear extinction.

Previous studies have shown that the PVT→CeA pathway is critical for the maintenance of fear memory[3,21]. And we demonstrated that the TRNrv neurons, which suppress dMTm neurons including PVT neurons (Fig. 4a), show increased firing activities during extinction learning (Fig. 3f, g). Therefore it is possible that the TRNrv may promote fear extinction by interrupting the maintenance of fear memory during extinction learning. This possibility is supported by our result showing that the silencing of dMTm→CeA circuit promotes fear extinction (Fig. 5).

A previous study has shown that the PVT, which is a part of the dMTm, does not encode prediction error in aversive learning[22]. Thus the facilitation of the extinction by TRNrv→dMTm pathway is unlikely to be related to the negative prediction errors generated by the conditioned tones. Rather, it is more likely that the TRNrv dampens the salience of the aversive tone encoded by the PVT[22], which might allow safety information to be encoded better in other extinction-related circuits—for example, the infralimbic cortex (IL)→the thalamic nucleus reuniens (RE) pathway[5].

We consistently observed elevated freezing levels in retrieval test either by inhibiting the somas of TRNrv neurons (Fig. 2f,

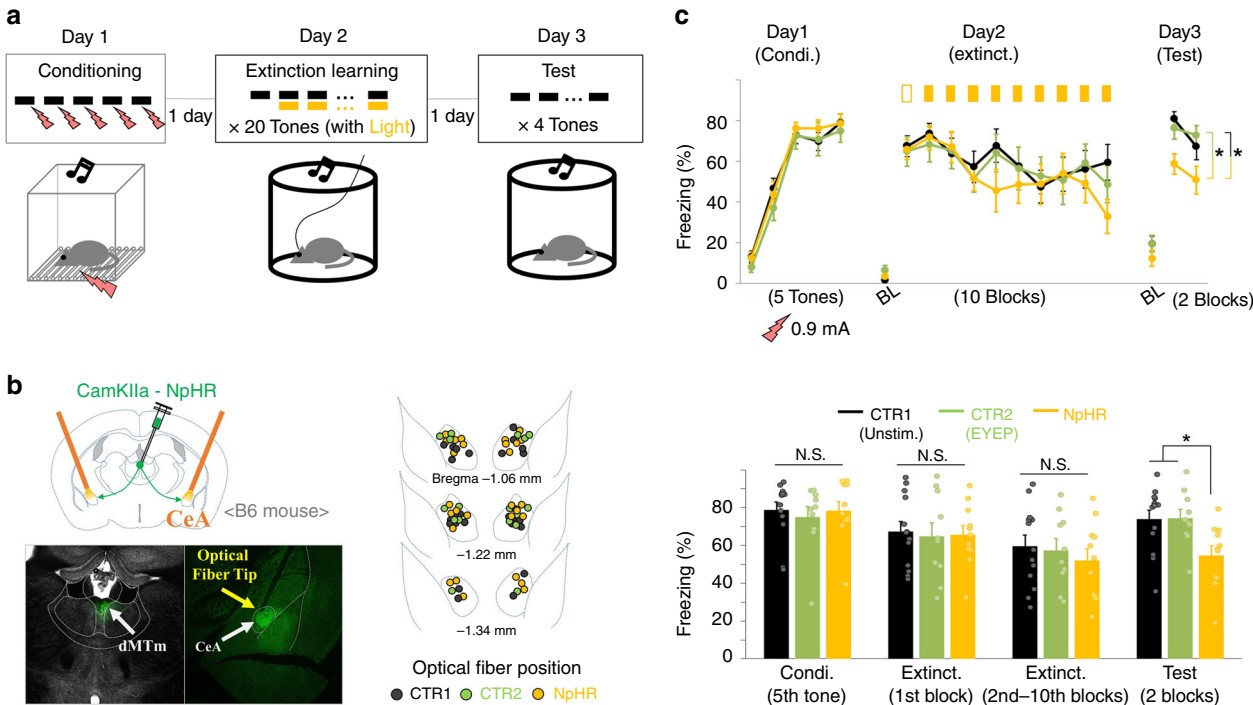

**Fig. 5** The suppression of dMTm→CeA pathway during extinction learning promotes fear extinction. **a** Experimental protocol. **b** Left, Schematic depiction for optogenetic inhibition of dMTm→CeA pathway. Right, The positions of fiber tips are marked by yellow dots for the stimulated group, gray dots for the unstimulated control group (CTR1, Unstim.), and green dots for the inactive-virus control group (CTR2, EYFP). **c** Top panel, Optogenetic inhibition (561 nm light, continuous pulse during the tone) of the dMTm→CeA pathway during extinction learning ($n = 13$ mice for CTR1 (Unstim.), $n = 10$ mice for CTR2 (EYFP), and $n = 11$ mice for stimulated group) induced the reduction of freezing levels during the retrieval test (four tones of the test day, two-way RM ANOVA followed by Holm–Sidak method, $F_{(2, 31)} = 5.214$, $P = 0.011$; Unstim. vs. NpHR, $t = 2.860$, $P = 0.022$; EYFP vs. NpHR, $t = 2.744$, $P = 0.02$; Unstim. vs. EYFP, $t = 0.0655, 0.948$) but did not affect freezing levels during extinction learning (3rd–20th tones of Extinct., two-way RM ANOVA, $F_{(2, 31)} = 0.423$, $P = 0.659$). No significant difference between the baseline (BL) freezing levels of the three groups was observed in extinction learning (Day 2, Kruskal–Wallis one-way ANOVA on Ranks, $H_{(2)} = 3.481$, $P = 0.175$) and retrieval test (Day 3, Kruskal–Wallis one-way ANOVA on Ranks, $H_{(2)} = 2.756$, $P = 0.252$). Bottom panel, Quantified data of the top panel. One block is the average of two tone trials. All data are presented as mean ± SEM. N.S., not significant. *$P < 0.05$. See Supplementary Table 1 for values of post hoc test

Day 3) or by specifically inhibiting the TRNrv→dMTm pathway (Fig. 4c, Day 3). However, we observed inconsistent results in freezing levels during extinction learning (Day 2): the elevated freezing level by inhibiting TRNrv somas (Fig. 2f, Day 2) but intact freezing level by inhibiting the TRNrv→dMTm pathway (Fig. 4c, Day 2). This might be because the inhibition of TRNrv somas includes the inhibition of TRNrv projections to other limbic thalamus than the dMTm. Along the boundary between the PVT/central medial thalamic nucleus and the MD, there is a thin area called "transition zone"[23]. The neurons in this area are known to project to the basolateral amygdala (BLA)[23], which is known to be important for fear expression[24,25]. We also observed the axonal terminals of TRNrv PV neurons in the "transition zone" (Fig. 1d, medial boundary of the MD), indicating that the inhibition of TRNrv somas may affect TRNrv→"transition zone" as well as the TRNrv→dMTm pathway, thus it may affect fear expression level as shown in Fig. 2f, Day 2. Nonetheless, physiological and functional assessments of "transition zone"→BLA circuit remain to be further investigated in the future.

A previous histological study showed that the IL projects to TRNrv[26]. Likewise, there are functional similarities between the IL and the TRNrv. Specifically, a previous study reported that optogenetic inhibition of the IL during extinction learning left within-session extinction intact but impaired subsequent retrieval of extinction[27]. Our TRNrv→dMTm inhibition yielded the same result (Fig. 4c). Also, it has been shown that the IL becomes

responsive to the conditioned tone at 24 h after the conditioning[28] as we observed in the TRNrv (Fig. 3f, g). In fact, this IL has been intensively implicated in fear extinction by electrical[28,29], pharmacological[24,30], and optogenetic approaches[27]. Therefore, considering the anatomical connection and the functional relevance between the IL and the TRNrv, it is likely that the TRNrv receives inputs from the IL during extinction learning and may play a role as a converter to change the excitatory output of the IL into the inhibitory input to the fear center, i.e., the dMTm, to suppress fear signal, thus promoting fear extinction.

In the previous studies, the IL has been elucidated in associative learning with various types of cues—for example, the light[31,32] or even a contextual cue[33,34]. This suggests that the TRNrv might be also responsible for different forms of cues other than the auditory tone, as the IL is.

Interestingly, a recent study showed that the pharmacological inhibition of the RE, which is positioned in the ventral part of the midline thalamus, impairs fear extinction[5]. It has been shown that the RE receives input from the IL[35]. Considering that the IL also project to the TRNrv, it seems that the IL simultaneously signals two different parts of the midline thalamus to promote fear extinction: (1) the ventral part, i.e., the RE, which would encode the information of safe context and prevent overgeneralization of conditioned fear[5,36] and (2) the dorsal part, i.e., the dMTm, of which suppression via the TRNrv would interfere the maintenance of fear memory[3,21].

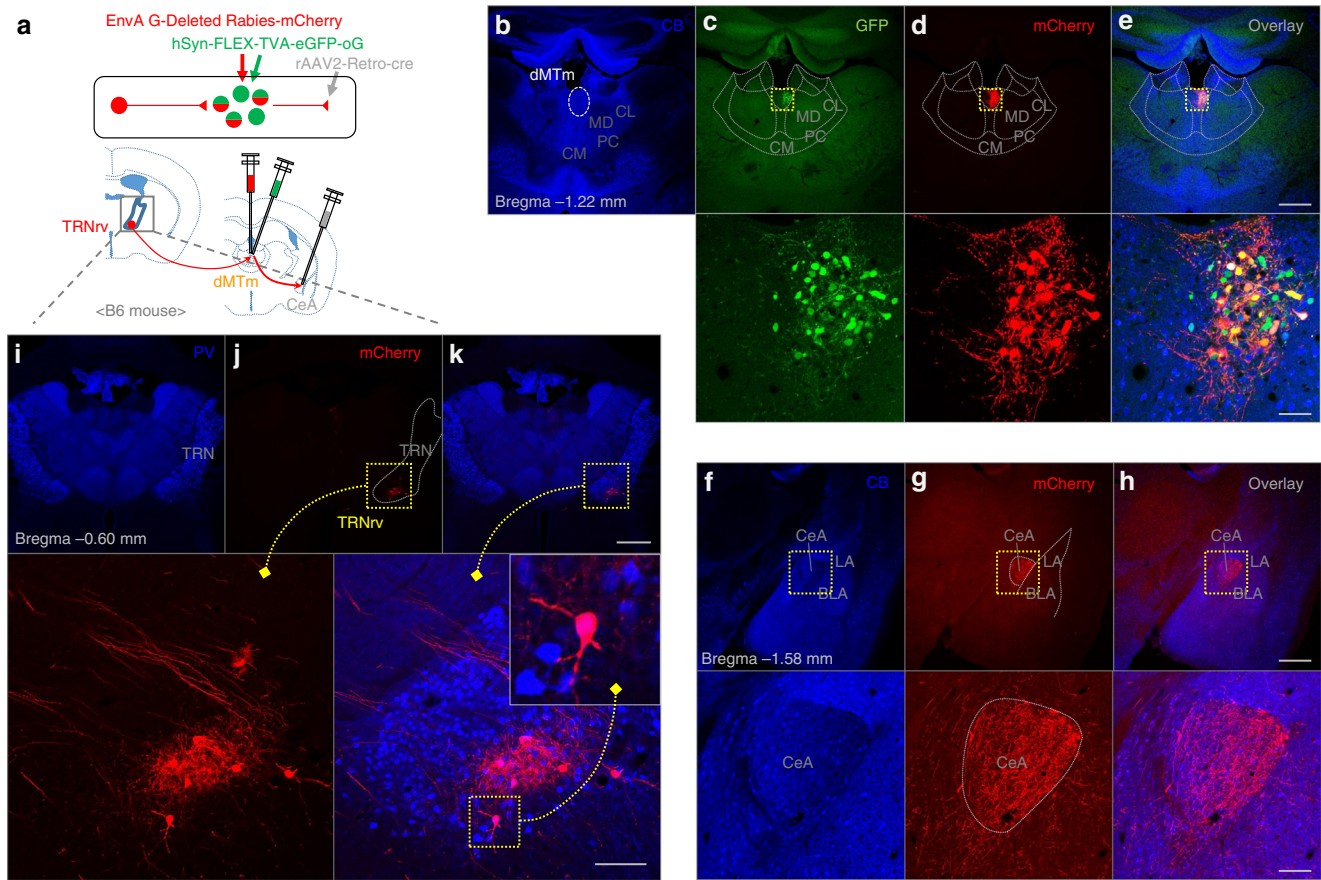

**Fig. 6** TRNrv neurons are disynaptically connected to the CeA through the dMTm. **a** Schematic depiction of virus injection. Retrograde-cre virus was injected into the CeA, which induces cre expressions in CeA-projecting dMTm neurons. Cre expression allows helper virus (green) to be expressed in the dMTm neurons, which allows expression of rabies virus (red) in the dMTm neurons. Finally, presynaptic neurons of the dMTm neurons show mCherry signal (red). **b** CB immunostaining delineates the boundaries of the limbic thalamus. **c** The expression of helper virus in the dMTm is shown. **d** The expression of rabies virus in the dMTm is shown. **e** Overlaid image. Scale bar, 500 μm. Magnified images. Scale bar, 50 μm. **f** CB immunostaining delineates the boundaries of the amygdala. Lower row, Magnified image of dotted square. **g** Restricted injection of retrograde-cre virus was confirmed by axon terminals of dMTm neurons in the CeA. **h** Overlaid image. Scale bar, 500, 100 μm. **i** PV immunostaining delineates the boundary of the TRN. **j** The expression of rabies virus in the TRNrv is shown and magnified image is shown in the lower panel. **k** Overlaid image. Scale bar, 1000 μm. Magnified images. Scale bar, 100 μm. LA lateral amygdala, BLA basolateral amygdala

Recent study reported that somatostatin (SOM) neurons in middle TRN receive inputs from the CeA[37], suggesting a possible role of TRN SOM neurons in the processing of emotional information. This study showed that, in middle TRN, around 15–60% of PV-positive neurons co-express SOM and, importantly, showed the co-expression of PV and SOM in rostral part of the TRN. These results raise a possibility that, in the TRNrv, the PV neurons that co-express SOM might be important for fear extinction.

## Methods

**Animals and surgery**. Animal care was provided and all experiments were conducted in accordance with the ethical guidelines of the Institutional Animal Care and Use Committee of the Korea Advanced Institute of Science and Technology and the Institute for Basic Science, Korea. Mice were maintained with free access to food and water under a 12-h light/dark cycle, with the light cycle beginning at 8:00 a.m. For all experiments, only male mice were used. For the injection of tracers or viruses, a custom-designed elongated (Sutter Instrument CO., P-87) borosilicate pipette (Final ID: 20–40 μm, World Precision Instruments, INC., 1B120F-3) was used. No statistical methods were used to predetermine sample size. No randomization was used to allocate the samples to experimental groups, and the investigators were not blinded to the allocation during experiments.

For the anterograde tracing surgery, B6.Pvalb-IRES-Cre (The Jackson Laboratory, no. 008069) mice aged 8–11 weeks were placed in a stereotaxic device (David Kopf Instruments) under ketamine/xylazine (75 and 5 mg/kg, respectively) anesthesia. We used the active form of GFP virus (AAV9-EF1a-DIO-hChR2

(H134R)-eYFP.WPRE.hGH, Addgene 20298) for tracing experiments because it shows strong axonal fluorescence. This GFP virus was pressure injected (Parker Hannifin Corp., Picospritzer III) into the TRN (anteroposterior/mediolateral/dorsoventral (AP/ML/DV), −0.6/1.4/3.6 mm). The mice were sacrificed for histological examination 3 weeks after the surgery.

For the retrograde tracing surgery, C57BL/6J mice aged 8–11 weeks were placed in the stereotaxic device under ketamine/xylazine (75 and 5 mg/kg, respectively) anesthesia. The retrograde tracers CTB (0.5% diluted in distilled water; List Biological) or fluorogold (FG; 2% in 0.1 M cacodylate buffer; Fluorochrome) were iontophoretically injected (for CTB, 7/7 s on/off duty cycle, 1 μA; for FG, 2/2 s on/off duty cycle, 1 μA, for 3 min) into the following brain regions (coordinates relative to Bregma): dMTm (AP/ML/DV, −1.34/0/3.0 mm), dMTl (AP/ML/DV, 1.34/1.0/3.0 mm). The mice were sacrificed for histological examination 5 days after the surgery.

For the rabies virus surgery, C57BL/6J mice aged 8–11 weeks were placed in a stereotaxic device (David Kopf Instruments) under ketamine/xylazine (75 and 5 mg/kg, respectively) anesthesia. The rAAV2-retro-CAG-Cre virus (UNC Vector Core, [rAAV2-Retro virus: AAV-CAG-Cre, Serotype: rAAV2-Retro]) and helper virus (AAV8-hSyn-FLEX-TVA-P2A-eGFP-2A-oG, UNC Vector Core) were pressure injected (Parker Hannifin Corp., Picospritzer III) into the CeA (AP/ML/DV, −1.22/2.5/4.3 mm) and the dMTm (AP/ML/DV, −1.34/0/3.0 mm), respectively. After 8 days, rabies virus (EnvA G-Deleted Rabies-mCherry, UNC Vector Core) was injected into the dMTm (AP/ML/DV, −1.34/0/3.0 mm). After 11 days, the mice were sacrificed for histological examination.

For optogenetic behavior in Figs. 2 and 4, Pvalb-IRES-Cre mice on a B6/129 F1 background were produced by mating B6.Pvalb-IRES-Cre (The Jackson Laboratory, no. 008069) mice with 129 S4/SvJae (The Jackson Laboratory, no. 009104) mice. For optogenetic behavior in Fig. 5, we used C57BL/6J (B6) (The Jackson Laboratory). These B6.129.PV-Cre mice (8 weeks old) and B6 mice were

placed in the stereotaxic device following the administration of ketamine/xylazine (75 and 5 mg/kg, respectively). Custom-generated ChR2 (see the "Virus" subsection below) for Fig. 2, AAV9.CBA.Flex.Arch-GFP.WPRE.SV40 (Addgene 22222) for Figs. 2 and 4, and rAAV5/CamkII-eNPHR3.0-EYFP-WPRE-PA, rAAV5/CamkIIa-EYFP for Fig. 5 was injected (0.2–0.3 μL) using pressure (Parker Hannifin Corp., Picospritzer III) into the TRNrd (AP/ML/DV, −0.6/1.4/3.2 mm), the TRNrv (AP/ML/DV, −0.6/1.0/4 mm), or the dMTm (AP/ML/DV, −1.34/0/3.0 mm). The injection pipette was removed slowly after a diffusion period of 10 min, then the optical fiber (Doric Lenses Inc., 100 μm core, 0.22 NA, ZF 1.25, DFL) was implanted with opaque dental cement. The dental cement was mixed with black powder (Art-Time, Tempera paint powder) to prevent the light leakage. The animals were allowed 3 weeks for complete recovery from the surgical procedure and for virus expression.

For the in vivo recordings in Figs. 3 and 4 and Supplementary Figs. 6 and 7, the virus injection procedures and the surgical implantation of the tetrodes were performed under ketamine/xylazine (75 and 5 mg/kg, respectively) anesthesia in B6.129.PV-Cre mice (8 weeks old). After the injection of custom-generated ChR2 (see the "Virus" subsection below) into the TRNrv (AP/ML/DV, −0.6/1.0/4 mm), the microdrive containing four tetrodes (16 channels; Neuralynx, Inc., Harlan 4 Drive) and the optical fiber (Doric Lenses Inc., 100 μm core, 0.22 NA, ZF 1.25, DFL) were inserted into the TRNrv (AP/ML/DV, −0.6/1.0/4 mm) or the dMTm (AP/ML/DV, −1.34/0/3.0 mm). For Supplementary Fig. 3b–d, the tetrodes and the optical fiber were implanted into the trunk region of primary somatosensory cortex (S1Tr, AP/ML/DV, −1.46/1.6/−0.5 mm) of B6.129.PV-Cre mice (8 weeks old). The optical fiber were closely located (~0.5 mm) to the tetrodes with a visual inspection.

A stainless steel screw was fixed in the skull over the right prefrontal cortex (AP/ML, 1.5/1.5 mm) or the cerebellum (AP/ML, 5.0/0.0 mm), and an uncoated stainless steel wire of the microdrive was tied to the screw as a ground or a reference for the tetrodes. Dental cement was applied to fix in place the microdrive, optical fiber, and stainless steel wires. All mice were housed singly to preserve the optical fiber and keep the microdrive intact. The animals were given 3 weeks to allow for complete recovery from the surgical procedure and for virus expression.

**Virus.** Channelrhodopsin fused with superfolder GFP (ChR2-sfGFP) was designed and synthesized from published sequences using codon optimization for *Mus musculus* (DNA2.0). To express ChR2-sfGFP in the Pvalb-IRES-Cre mouse, the faithful flexed AAV vector under the control of the human synapsin promoter (aavSyn-Jx) was generated using a PCR-amplified human synapsin promoter and lox66/lox7 sites[38]. For the Cre-dependent switch "on" version (aavSyn-Jx-rev-ChR2-sfGFP), ChR2-sfGFP was reversely inserted into the aavSyn-Jx via the HindIII and EcoRV restriction-enzyme sites. The viruses were produced with serotype 1 or 7 and purified using cesium chloride gradients[39]. For optogenetic inhibition, the AAV9.CBA.Flex.Arch-GFP.WPRE.SV40 (Addgene 22222) virus was used. The reason we utilized the Arch virus was that it provides optimal terminal expression and has been shown to inhibit neurotransmitter release[40–42]. Although a recent study reported that terminal inhibition using an Arch virus paradoxically enhanced terminal release[40], it was only apparent after long-duration illumination; we used short-duration illumination (30 s), which does not significantly increase spontaneous neurotransmitter release as this previous study has shown. The retrograde CAV-2-Cre virus was purchased from the Montpellier Vector Platform, France (titer ~2.5 × 10E12 pp/mL).

**Behavior.** All behavioral experiments were conducted under white noise (65–70 dB) presentation. For the fear conditioning experiments, mice were placed into a metallic rectangular chamber (Context A) with a surface grid connected to an electrical shocker (Coulbourn Instruments) housed inside a sound-attenuating box (Coulbourn Instruments). The mice were fear-conditioned by three or five presentations (at an interval of 120 s) of a tone (3 kHz, 30 s, 80 dB) that was co-terminated with electric foot shocks (Intensity, 0.3, 0.5, 0.7, or 0.9 mA as designated in the figures; duration, 1 s). Mice remained in the chamber for 60 s after the last tone+shock presentation and then were returned to their home cages. After 24 h, mice were placed in a cylindrical acrylic box (Context B) to receive extinction learning, in which the mice was exposed to the tones (at random intervals of 30–60 s) without electric shock. During extinction learning on Day 2, in the first tone for Figs. 2 and 4 or first two tones for Fig. 5, both the control and stimulated groups were free from optogenetic stimulation to confirm proper fear memory acquisitions, and only the stimulated group were stimulated by light stimulation during the rest of tone presentations. After the last tone was presented, the mice were returned to their home cages. For the retrieval test conducted 24 h later, the mice were exposed to four tones without light stimulation in the same environment where the extinction learning took place (Context B). The video recording, tone presentation, and light stimulation were synchronized using the FreezeFrame software (Coulbourn Instruments) and the PulsePal[43] stimulator (http://www.open-ephys.org/pulsepal/).

We performed Elevated Plus Maze to determine their anxiety level. Mice were placed in a plus maze for 5 min, consisting of two opposing open arms (each 30 × 5 cm) and two opposing closed arms (each 30 × 5 cm) with 15-cm-high walls, elevated to 30 cm above floor level, while the mice were connected to the optical patch cord for optogenetic stimulation.

We performed Open Field Test to test locomotor activity. We placed mice in the central region (a square of 20 × 20 cm) of an open field box (40 × 40 × 40 cm) and analyzed the locomotion over 30 min using the EthoVision software.

**Optogenetic stimulation.** To deliver the light, the implanted optical fiber (Doric Lenses Inc., 100 μm core, 0.22 NA, ZF 1.25, DFL) was connected to the optical patch cord (Doric Lenses Inc., MFP_100/125/900-0.22_2m_FC-ZF1.25 with flange) through the sleeve (Doric Lenses Inc., SLEEVE_ZR_1.25). The patch cord was connected to a rotary joint (Thorlabs, Inc., RJP-Custom) to prevent the pressure by twisting. For the light source, a 450 nm blue laser (Changchun New Industries Optoelectronics Technology Co., MDL-III-450) or a 561 nm yellow laser (Changchun New Industries Optoelectronics Technology Co., MGL-FN-561) was used.

For the control groups in all optogenetic experiments except for CTR2 (EYFP) group in Fig. 5, we injected the same active virus as for the experimental group and blocked the light transmission into the brain. To control potential behavioral deficits caused by the light itself, we employed the following procedures: (1) We used opaque dental cement, which completely blocked the light reflection from the inside of the brain, and covered sleeve and tube curtain, which completely blocked the light leakage at the junction of the patch cord and the optical fiber (Supplementary Fig. 3a). (2) To be safe from the heating effect, we carefully chose the intensity of the light (140 mW/mm²) based on the previous study[44]. Also, we confirmed that our light stimulation does not cause heating effect by recording single units at various light intensities (Supplementary Fig. 3b–d).

For CTR2 (EYFP) group in Fig. 5, we injected CamKIIa-EYFP virus into the dMTm and implanted optical fibers into the CeA, then delivered the light during extinction learning.

The following optogenetic stimulations were accompanied by the tone during extinction learning: 1 or 10 Hz blue light stimulation (6.3 ms duration, 0.5 mW at the 100 micron fiber tip, which is converted to 64 mW/mm²) for ChR2-expressing group or continuous yellow light stimulation (0.9–1.1 mW at the 100 micron fiber tip, which is converted to 115–140 mW/mm²) for the Arch-expressing group. The light intensity was measured by digital optical power meter (Thorlab, Inc., PM100D) before the beginning of the experiments.

**Histology.** Perfusions were performed under ketamine/xylazine (75 and 5 mg/kg, respectively) anesthesia. The animals were perfused first with saline (0.9%) and then with 4% paraformaldehyde (Tech & Innovation, BPP-9004) solution (100 mL). Brains were removed and cut into 50-μm-thick coronal sections with a vibratome (Leica, VT1200S). Sections were washed with phosphate buffer (0.1 M) and then treated with a blocking solution containing 3% normal donkey serum (Millipore, S30-100ML) and 0.2% Triton X (Sigma, T8787) for 40 min at room temperature. The following primary antibodies diluted in phosphate buffer containing 0.1% normal donkey serum and 0.1% Triton X were used: anti-cholera toxin-B subunit (goat, 1:20,000–30,000; List Biological, 703), anti-fluorogold (rabbit, 1:10,000–20,000; Fluorochrome), anti-PV (mouse, 1:3,000–5,000; Swant, 235), and anti-calbindin (mouse, 1:3,000–5,000; Swant, 6B3). After primary antibody incubation (1 day at room temperature or 2–3 days at 4 °C), the sections were treated with secondary antibodies labeled with fluorescent dye (Alexa 488, Cy3, or Cy5; 1:500, 2 h at room temperature; Jackson ImmunoResearch). Sections with fluorescent staining were mounted with Vectashield containing 4′,6-diamidino-2-phenylindole (Vector Laboratories, H-1400). Photographs were captured using either a microscope (Nikon) or a confocal laser scanning system (Nikon).

**Behavioral data analysis.** For fear conditioning experiments, the freezing levels of the mice were automatically measured by FreezeFrame software. For fear extinction and retrieval tests, because the movements of the lines above the head of the mouse would interfere with the reliable analysis of the software, the freezing levels (defined as behavioral arrest except for movements associated with respiration) were scored by an observer viewing the video recording. For this analysis, the videos were given randomized numbers to blind the investigator to the treatment condition. During the analysis, a small percentage of mice (5.85%, 12 out of 205 mice) were excluded because they did not exceed the pre-established criterion for acquisition of conditioned freezing ( > 10% freezing during the presentation of the first tone in extinction learning). No further exclusions were made. For the elevated plus-maze test, the percentage of open arm entries, which is the number of open arm entries divided by the total (open + closed) arm entries multiplied by 100, was scored by the software (EthoVision XT 11, Noldus Information Technology).

**Single unit recording.** In the single unit recording experiments, the microdrive was connected to the tether line with a 16-Channel analog head-stage amplifier (HS-16; unity gain, 1.00; Neuralynx Inc., USA) to record single unit activity. The recorded data were obtained at a sampling frequency of 32 kHz using a Digital Lynx DX64 A/D converter. Signal acquisition and recordings were performed using the Cheetah software (version 6.5; Neuralynx Inc., USA). The bandpass filter used for single unit recording was 600–6000 Hz. The threshold for spike detection was 50–60 μV. Spike sorting was performed using MClust 3.5 (A. David Redish, http://redishlab.neuroscience.umn.edu) in Matlab (The MathWorks, Inc.). The sorted units of which the violation of the inter-spike interval (<2 ms) was <1.1% were used

for the analysis. If the response of a neuron to the optical stimulation satisfied the following criteria, it was classified as PP neuron: if firing rate at any point from 0 to 8 ms after the optical stimulation is over than a $Z$-score of 3.72 or firing rate is continuously over than a $Z$-score of 1.96 during 8 ms. For the firing responses to tones, the data were normalized to the baseline period, which was 5 s before each tone. Electrode positions were confirmed by postmortem histological examination. After the experiments, a micro-lesion was made by applying anodal current. To precisely distinguish the position of each of the four tetrodes, a current of 40 µA for 10 s was applied to one tetrode and a different current was applied to the other three tetrodes (20 µA for 10 s), but for two tetrodes, the current was applied two or three times while the tetrode was moved up or down with 300-µm intervals to differentiate the lesioned sites. For Supplementary Fig. 3b–d, 1 week after the surgery, the mice were located in home cage and the recording was started when the mice were resting. The baseline data were recorded for 30 s, then the recording for 30 s with light stimulation was performed. This procedure was repeated for each light intensity with 30-s intervals.

For fear behavior, 3 weeks after the surgery, the mice were connected to the recording device for cell-hunting. For 5 days, the electrodes were lowered by approximately 20 µm until they reached the position in which the number of detected single units was at a maximum. Two days before fear conditioning, the single unit responses to four presentations of neutral tone were measured while the mouse was in the cylindrical acrylic box (Context B, matte white). To identify the tagged neurons, brief 10 Hz blue light stimulation was applied (30 s, 6.3 ms duration, 4 mW at the fiber tip, which is converted to 510 mW/mm² for a 100-micron fiber). After a 5-min rest, the first neutral tone was given. No optical stimulation was applied during presentations of the neutral tone. Two days later, the fear conditioning experiments were performed. One day after the fear conditioning, the single unit responses to 18 presentations of conditioned tone during the extinction learning were measured while the mouse was in the cylindrical acrylic box (Context B, matte white). The same tagging protocol was applied 5 min before the first tone of extinction learning. For Supplementary Fig. 6c, d, the data were acquired from second through to the last extinction tones, because, in the first tone, no light stimulation was delivered to either the control group or the stimulated group as in the behavior test (Fig. 2d).

**Statistical analysis**. The statistical analysis was performed using the commercially available software (SigmaPlot 12.0, Systat Software, Inc.). For all analyses, the tests for normality and equal variance were automatically performed by the software to appropriately select parametric or nonparametric test methods, and the post hoc analyses used were those automatically suggested by the software.

**Reporting summary**. Further information on research design is available in the Nature Research Reporting Summary linked to this article.

## Data availability
The data sets that support the findings of this study are available from the corresponding author.

## Code availability
Matlab code used in this project for data analysis is available from the correspondence author upon reasonable request.

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

## Acknowledgements

We thank S. Keum and B. Lee for insightful discussion. We thank S.J. Kim and J. Baek for technical assistance of tetrode manufacturing. This work was supported by a grant from IBS (IBS R001-D1) to H.-S.S. and KIST institutional program (2E26860) to J.K.

## Author contributions

J.-H.L. and H.-S.S designed the experiments. J.-H.L. performed and analyzed the anatomical studies, performed and analyzed the in vivo electrophysiological studies, and visualized the figures. J.-H.L., C.-F.V.L., J.P. and J.J. performed and analyzed the behavioral studies. J.K. provided ChR2 virus. J.-H.L., K.-H.L. and H.-S.S. wrote the paper.

## Competing interests

The authors declare no competing interests.
