## [Peer Review File · Nature Communications]

Reviewers' Comments:

Reviewer #1:

Remarks to the Author:

In the present work, the authors reported very interesting results regarding the role of the thalamic reticular nucleus (TRN) in fear extinction. They were able to show that the rostroventral part of the TRN (TRNrv) projects to the dorsal midline thalamus (dMT), which is known to be involved in the retrieval of fear learned responses. The authors applied optogenetic excitation and inhibition in the TRNrv and showed facilitation and inhibition, respectively, of fear extinction. They were able to see that the TRNrv projection suppressed the spiking activity of the dMT and that TRNrv neurons selectively respond to conditioned but not neutral stimuli. As much as the experiments were carefully done and the results are interesting, there are a number of questions that the authors should address. In the experiments shown in Figure 2, the levels of freezing for the control group of optogenetic stimulation of the TRNrv, shown in panel d, are much higher than for the control group of optogenetic inhibition of the TRNrv, shown in panel f. This fact is particularly problematic because the authors are reporting that TRNrv stimulation induces an increase in fear extinction with controls showing high levels of freezing, whereas the TRNrv inhibition induces a decrease in fear extinction with controls showing low levels of freezing. In fact, the levels of freezing in the control group of Fig 2 f is close to the one shown in Fig 2 d after TRNrv optogenetic stimulation.

It is not clear why inhibition of presynaptic terminals from the TRNrv in the dMT decreases fear extinction during retrieval and not during the learning phase, as opposed to the inhibition of the TRNrv neurons. The authors claim that "...this might be because the inhibition of TRNrv soma includes the inhibition of TRNrv projections to other limbic thalamus than the dMTm..." such as an area between the dMT and the MD (transition zone) that projects to the basolateral amygdala which is important for fear expression. If this is the case, we would expect that optogenetic manipulation of the dorsal part of the TRN, which, as shown in this report, projects to this transition zone, would also influence fear extinction. Moreover, the use of archaerhodopsin as an inhibitory optogenetic tool of presynaptic terminals is problematic because after the inhibition of the presynaptic terminals there are spontaneous post-discharges associated with increased neurotransmitter release. How did the authors try to circumvent this methodological problem?

The authors were able to show that TRNrv neurons selectively respond to conditioned but not neutral stimuli. However, they examined only auditory tones. Does it apply to other sensory stimuli, such as light, or even context? This information would be important to expand the TRNrv's role in the control of fear extinction to other forms of conditioned stimuli and, perhaps, general contextual clues. To this end, it seems to be critical to understand the main afferent sources of input to the TRNrv. The examination of the neural inputs to the TRNrv would give important cues related to how this region articulates with other circuits related to fear conditioning and how the TRNrv neurons would be selectively responsive to conditioned but not neutral stimuli.

Reviewer #2:

Remarks to the Author:

This study by Lee and colleagues investigates the role of reticular nucleus of thalamus (TRN) in fear extinction. TRN is generally considered to be a major extrinsic source of inhibition and dis-inhibitions in thalamus. Through a series of retrograde tracing, electrophysiological and optogenetic studies, the authors show that TRN projections to the dorsal mid-line thalamus (with CeA as a potential downstream target) to mediate its effects on fear extinction. This is a novel and potentially path-breaking study. However, some of the work is speculative and needs significant revisions before being

considered for publication.

MAJOR ISSUES:

- 1) In lines 3 and 4 of the Introduction, the authors mention that the limbic thalamus plays a critical role in fear extinction and cited three articles. However, two of those articles (citations #3&4) concluded the dorsal mid-line (dMT) thalamus is not involved in fear extinction (rather it plays a role in fear expression). Animals with dMT inactivation show decreased freezing during extinction training, which is due to fear attenuation (see Figure 2 of Padilla-Coreano et al., 2012). However, a recent paper from Ramanathan and colleagues (2018; Nature Communications) has implicated the ventral mid-line nuclei in fear extinction. The authors should cite that work as evidence of mid-line thalamic involvement in fear extinction and correct the miscited role for dMT in extinction throughout the manuscript.
- 2) TRN nuclei are rich in GABAergic neurons. However, it is unclear what proportion of cells are PV+. In Fig 3 they mention that 24-32% of cells are light-responsive, which challenges the conclusion that roughly 1/3rd of TRN cells are PV+ cells. More importantly, do all of the projections from TRN to dMT originate from PV+ cells as opposed to other GABAergic cells? The authors should either 1) answer this question quantitatively or at least cite relevant articles that support their claim that dMT projectors are PV+ cells or 2) temper the language that these effects are mediated by PV+ neurons in TRN.
- 3) The behavioral controls for the optogenetic experiments are unclear. The authors should describe the procedures used in the control (CTR) group, which is currently not described in the relevant figures, legends, or methods. At minimum, the authors should include an inactive virus group that receives light stimulation to control for light-induced behavioral deficits in TRNrv.
- 4) It is interesting that authors observed deficits with 10 Hz, but not 1 Hz, stimulation. The authors should clarify the rationale for using two different optical stimulation procedures and explain why they observed with one stimulation but not the other.
- 5) In Fig 3F&G, the authors show changes in neuronal firing rates among TRN neurons in response to an auditory CS after fear conditioning. However, it is unclear how these changes relate to the hypothesized role for TRN in fear extinction. The work is not well connected to the other experiments.
- 6) The authors report anatomical data showing that TRNrv neurons disynaptically project to the central amygdala (CeA) via the dMT. On this basis they suggest that the TRNrv influences extinction via this projection, however there are no functional data to support this conclusion. Importantly, the dMT itself has not been implicated in fear extinction in previous work (see #1). Moreover, dMT projections to CeA have been shown to be involved in fear conditioning and retrieval (see Padilla-Coreano et al., 2012; Do-Monte et al., 2015; Penzo et al., 2015). Hence, the authors should either provide functional evidence to support the role for TRN->dMT->CeA projections in fear extinction or remove this part of the story.
- 7) The discussion is abrupt. The authors should edit this section to improve its flow, and most importantly, should discuss the implications for a role of TRN in fear extinction in the context of other work showing that the ventral midline nuclei (including the nucleus reuniens) are important for fear extinction.

MINOR ISSUES:

- 1) In the second paragraph of the methods section, the authors mention an active virus (with hCHR2) in a tracing study. Is this a typo?
- 2) In Supplemental Fig 2 the optic fiber tracts are not visible. The authors should provide a figure illustrating the localization of the optic fibers in TRN.
- 3) The authors should report the software used for statistical analyses and the relevant p-values in the results section (not just in the figure legends).
- 4) The article needs extensive proof-read to correct the extensive number of typos and grammatical and spelling errors.

Reviewer #3:

Remarks to the Author:

In this paper by Lee et al., a series of experiments was conducted to investigate the role of the reticular thalamic nucleus (TRN) in extinction of conditioned fear. Anatomical tracing studies were conducted to show that rostroventral TR (TRNrv) sends inhibitory projections to the dorsomedial thalamus (dMT), which in turn projects to amygdala. Exciting TRNrv enhances fear extinction, whereas inhibiting TRNrv neurons (or inhibiting TRNrv terminals in dMT) impairs fear extinction. TRNrv neurons respond to a tone CS after it has been paired with shock, but not before. The authors argue that this pattern of results indicates a prominent role for TRNrv in the extinction of conditioned fear.

These findings are timely and would be of wide interest to the neuroscience community, especially for researchers who study the neural basis for fear and anxiety. The study impressively integrates a range of modern tools for genetic targeting of neural pathways, optogenetic stimulation, and single-unit recording. However, there are a few major issues that need be addressed (see below). First and foremost, the freezing analyses do not include a comparison of tone-evoked responses against freezing in the absence of a tone. Another issue is that some of the reported effects appear to possibly arise from differences among control groups, rather than experimental groups.

GENERAL COMMENTS

A major issue is that throughout the paper, the extinction graphs show only freezing during the tone. There is no comparison against the pre-tone baseline. It is therefore not possible to tell whether the authors are measuring conditioned freezing evoked by the tone, or persistent mobility/immobility that is lasting throughout the entire extinction session (regardless of whether the tone is on or off). It is imperative that the authors compare freezing in the presence versus absence of the tone, to validate the claim that this is conditioned freezing evoked by the tone. A plot of freezing during the pre-CS period (or of the difference score between CS and pre-CS freezing) should be shown, along with appropriate statistics. Pre-CS freezing should be scored blindly in the same manner as freezing during the CS.

It is also somewhat troubling that the "accelerated" extinction in the ChR2 group (Fig. 2d, Supp Fig. 3b) appears to result not only from "acceleration" of extinction in the ChR2 group, but also to poor extinction in the control group. Moreover, the "impaired" extinction in the Arch group (Fig. 2f) is obtained by comparison against a control group which exhibits lower freezing levels than other control groups (especially those from the ChR2 experiments). It looks like the Arch group in Fig. 2f would not show impaired extinction if it were compared against the control group for ChR2, and it is not clear that the ChR2 group in Fig. 2d would show significantly enhanced extinction if it were compared against the control group for Arch in Fig. 2f. It is a bit of a red flag for me when the findings of a study appear to be attributable to variations in the control groups, rather than the experimental groups. The findings would be more convincing if the data from control groups were more consistent across experiments.

One large piece that seems to be missing from this story is what happens when TRN is manipulated during fear acquisition, rather than extinction. Would stimulation of TRNrv impair the acquisition of conditioned fear? The title of the paper implies that TRNrv plays some specialized or specific role in fear extinction. But we don't really know this unless acquisition experiments are performed for comparison.

SPECIFIC COMMENTS

The disynaptic tracing data in Fig.5 is quite beautiful, but how many mice has this pattern been observed in? It appears to be n=1 but this is not entirely clear. Replication in additional mice might be advisable. The sample size for the tracing data in Fig. 1 also seems rather small, but in that case, the problem is ameliorated by showing the reversed labeling for the retrograde tracers in the supplementary figures. 2-3 replications of the disynaptic labeling results would be advisable.

Are the data in Fig 2d and supplementary Figure 3b completely independent replications of the extinction experiment (no overlapping mice between these experiments)?

Supplementary figure 2 is intended to show positioning of optical fibers in TRNrv and TRNrd, and optogenetically induced c-fos expression. Blue dashed lines are used to show the outline of the fiber position, but there does not to be any fiber track in the outlined tissue. How was it determined that the fiber was indeed positioned in this location?

How many mice were the cell recordings shown in Fig. 4 obtained from? The cells counts are rather low. A histology figure showing the electrode tracks would help to support the claim that these cells were all recorded from DMTm.

The first 2-3 paragraphs of the introduction could do a better job of introducing background on the TRN, for example, that it contains GABAergic projection neurons that inhibit other thalamic nuclei.

p. 3: "Guided by this connection map we carried out circuit analysis and found that TRNrv neurons are activated during extinction learning, which leads to the suppression of the dMTm neurons, resulting in fear extinction." The last bit is an interpretation, not a finding, no?

p. 3: "As a result, we observed axonal signals across the dMT (Fig. 1d)." This is poorly worded and confusing. Better to say "After virus injections into TRN, we observed GFP labeled axons throughout the dMT (Fig. 1d)."

p. 4: "and with precise positioning of optical fiber (Fig. 1c, 1e and 1g)." It appears that this should say Fig. 2c, 2e, and 2g?

p. 6: "As a result, surprisingly, TRNrv neurons showed increased firing activity selectively to the conditioned tones (Fig. 3f, first row, ALL neurons) but not to the neutral tones (Fig. 3d, first row, ALL neurons)." The word "surprisingly" seems inappropriate here. Isn't this what the authors were expecting?

p. 8: "we examined that whether" should say "we examined whether"

In some places, the authors write "excitated" where they should write "excited" ("excitated" is not a word).

The summary table for statistics in the supplement only gives omnibus F values for the ANOVAs. It would be nice to see the main and interaction effects as well.

p. 9: "Considering the well-known factor that the sensory sector of the TRN intimately communicates not only with sensory thalamus but also with corresponding sensory cortex 16, which cortical area could correspond to TRNrv-dMTm circuit?" This question is poorly worded and hard to understand. I assume that the authors are proposing future research to investigate which cortical regions are reciprocally connected to the TRNrv-dMTm circuit, since each thalamic region tends to be mated with

particular cortical areas. But the wording of this entire paragraph is confusing and awkward. Later in the paragraph, it is stated "it may be reasonable to select the infralimbic cortex as upstream brain area of the TRNrv," as if it is up to the authors to choose which cortical regions are connected to TRNrv-dMTm. This paragraph should be rewritten.

The discussion should address question about what functional role the TRNrv-dMTm circuit might play in extinction. Does this circuit simply inhibit fear and thereby contribute to a sense of safety? Is it involved in generating negative prediction errors that might drive extinction learning? Are there experiments that might be done to dissociate between these and other possibilities?

Responses to the Reviewers

(NCOMMS-18-36301)

“The rostroventral part of the thalamic reticular nucleus modulates fear extinction”

We would like to thank the editor and all three reviewers for their dedication to review our study and provide constructive suggestions. We have performed new experiments and have revised the manuscript carefully addressing each of the reviewers' comments as follows. We have underlined the changes in the manuscript with red color and have underlined the text in this response letter with blue color where we added new experiments.

Reviewer #1 (Remarks to the Author):

In the present work, the authors reported very interesting results regarding the role of the thalamic reticular nucleus (TRN) in fear extinction. They were able to show that the rostroventral part of the TRN (TRNrv) projects to the dorsal midline thalamus (dMT), which is known to be involved in the retrieval of fear learned responses. The authors applied optogenetic excitation and inhibition in the TRNrv and showed facilitation and inhibition, respectively, of fear extinction. They were able to see that the TRNrv projection suppressed the spiking activity of the dMT and that TRNrv neurons selectively respond to conditioned but not neutral stimuli. As much as the experiments were carefully done and the results are interesting, there a number of question that the authors should address.

Major issue 1:

In the experiments shown in Figure 2, the levels of freezing for the control group of optogenetic stimulation of the TRNrv, shown in panel d, are much higher than for the control group of optogenetic inhibition of the TRNrv, shown in panel f. This fact is particularly problematic because the authors are reporting that TRNrv stimulation induces an increase in fear extinction with controls showing high levels of freezing, whereas the TRNrv inhibition induces a decrease in fear extinction with controls showing low levels of freezing. In fact, the levels of freezing in the control group of Fig 2 f is close to the one shown in Fig 2 d after TRNrv optogenetic stimulation.

The reviewer pointed out that the freezing levels of the control groups are different. In the experiment of optogenetic excitation (Fig. 2d), we delivered a stronger shock (0.5mA) compared to the 0.3mA shock used in the inhibition (Fig. 2f and 2h) as designated in the figure (at the bottom of the Day1 in each Fig. 2d, 2f and 2h). The reason we applied the stronger shock in the excitation experiment was to avoid floor effect, allowing clear detection of freezing reduction, as previous study employed¹. That is, because we observed that 0.3mA shock induces mild freezing level during extinction learning and retrieval test, we increased intensity of shock to see clear reduction in freezing levels. For similar reason, we applied a weaker shock (0.3mA) in inhibition experiment (Fig. 2f and 2h) to avoid ceiling effect¹. We now describe this in the main text (**Page 6, Line 123-127**). In addition, to clearly show how the intensity of shock affects freezing levels in our setup, we performed new experiments with two different shock intensities (0.3mA and 0.7mA) (Response Fig. 1). We used the mice of same background (B6/129 F1) as used in the previous optogenetic experiments (Fig. 2). As expected, the mice with a stronger shock showed significantly increased freezing levels throughout the extinction learning on Day2 and the retrieval test on Day3 (Response Fig. 1). We added this data to **Supplementary Fig. 2** and revised the manuscript (**Page 6, Line 124-125**).

Response Fig. 1. The effect of different shock intensity in fear conditioning to the freezing levels during extinction learning and retrieval test.

Major issue 2:

It is not clear why inhibition of presynaptic terminals from the TRNrv in the dMT decreases fear extinction during retrieval and not during the learning phase, as opposed to the inhibition of the TRNrv neurons. The authors claim that "...this might be because the inhibition of TRNrv soma includes the inhibition of TRNrv projections to other limbic thalamus than the dMTm..." such as an area between the dMT and the MD (transition zone) that projects to the basolateral amygdala which is important for fear expression. If this is the case, we would expect that optogenetic manipulation of the dorsal part of the TRN, which, as shown in this report, projects to this transition zone, would also influence fear extinction.

One thing we want to clarify is that the TRNrd does not project to the transition zone. As we described in the result (Page 4, Line 86-87), the TRNrd projects to the dMTl which is on the lateral side of the MD, whereas the transition zone is on the medial side of the MD (Response Fig. 2, indicated by asterisk). For this reason, the inhibition of the TRNrd would not affect transition zone and we believe this is the reason why the manipulation of the TRNrd did not affect fear expression during fear extinction learning.

Response Fig. 2. (Originally from Fig. 2. of Mátyás et al., 2013)². *transition zone.

Major issue 3:

Moreover, the use of archaerhodopsin as an inhibitory optogenetic tool of presynaptic terminals is problematic because after the inhibition of the presynaptic terminals there are spontaneous post-discharges associated with increased neurotransmitter release. How did the authors try to circumvent this methodological problem?

The reviewer pointed out the possibility of increased neurotransmitter release when using archaerhodopsin in our study. Indeed, the previous study reported that sustained archaerhodopsin activation paradoxically enhanced spontaneous release³. In this report, the enhanced terminal release was only apparent after long-duration illumination; we used short-duration illumination (30s), which does not significantly increase spontaneous neurotransmitter release as this previous study has shown. We have described this in the method (Page 16, Line 364-368).

Major issue 4:

The authors were able to show that TRNrv neurons selectively respond to conditioned but not neutral stimuli. However, they examined only auditory tones. Does it apply to other sensory stimuli, such as light, or even context? This information would be important to expand the TRNrv's role in the control of fear extinction to other forms of conditioned stimuli and, perhaps, general contextual clues. To this end, it seems to be critical to understand the main afferent sources of input to the TRNrv. The examination of the neural inputs to the TRNrv would give important cues related to how this region articulates with other circuits related to fear conditioning and how the TRNrv neurons would be selectively responsive to conditioned but not neutral stimuli.

We thank the reviewer for this insightful comment. The reviewer pointed out the importance of checking neural inputs to the TRNrv to get the clues whether the TRNrv might respond to other forms of stimuli and to understand how the TRNrv articulates with other brain area. The previous study already reported that the TRNrv receives inputs from the infralimbic cortex⁴ (IL). In addition to this anatomical clue, there is an evidence for functional relevance between the IL and the TRNrv. A previous study showed that the optogenetic inhibition of the IL during extinction learning induces intact freezing levels during extinction learning but reduced freezing levels during retrieval test⁵. We observed exactly the same result by inhibiting the TRNrv→dMTm pathway (Fig. 4c). These previous studies raise a possibility that the selective response of the TRNrv to the conditioned stimuli could be related to IL activity. In fact, a previous study reported that the IL becomes responsive to the conditioned tone at 24h after the conditioning⁶, which is consistent with our result which shows the

increased response of TRNrv neurons at 24h after the conditioning (Fig. 3f and 3g). In addition, this IL activity has been shown to be critical for fear extinction by electrical^{6,7}, pharmacological^{8,9} and optogenetic approaches⁵. Therefore, considering the anatomical connection and the functional relevance between the infralimbic cortex and the TRNrv, it is likely that the TRNrv receives inputs from the IL during extinction learning and may play a role as a converter to change the excitatory output of the IL into the inhibitory input to the dMTm to suppress the fear signal. Also, the IL has been elucidated in associative learning by using various cues including the light^{10,11} and even contextual cue^{12,13}. Thus, the TRNrv might respond to different forms of cues as the IL does. We added this issue in the discussion (**Page 12, Line 265-278**).

Reviewer #2 (Remarks to the Author):

This study by Lee and colleagues investigates the role of reticular nucleus of thalamus (TRN) in fear extinction. TRN is generally considered to be a major extrinsic source of inhibition and dis-inhibitions in thalamus. Through a series of retrograde tracing, electrophysiological and optogenetic studies, the authors show that TRN projections to the dorsal mid-line thalamus (with CeA as a potential downstream target) to mediate its effects on fear extinction. This is a novel and potentially path-breaking study. However, some of the work is speculative and needs significant revisions before being considered for publication.

Major issue 1:

In lines 3 and 4 of the Introduction, the authors mention that the limbic thalamus plays a critical role in fear extinction and cited three articles. However, two of those articles (citations #3&4) concluded the dorsal mid-line (dMT) thalamus is not involved in fear extinction (rather it plays a role in fear expression). Animals with dMT inactivation show decreased freezing during extinction training, which is due to fear attenuation (see Figure 2 of Padilla-Coreano et al., 2012). However, a recent paper from Ramanathan and colleagues (2018; Nature Communications) has implicated the ventral mid-line nuclei in fear extinction. The authors should cite that work as evidence of mid-line thalamic involvement in fear extinction and correct the miscited role for dMT in extinction throughout the manuscript.

We thank the reviewer for this important comment. The reviewer correctly pointed out the miscitation of the study (Padilla-Coreano et al., 2012)¹⁴, in which it was concluded that the dMT was not involved in fear extinction. We corrected the miscitation in the manuscript (**Page 2, Line 42-44**). Regarding the other previous study (Do-Monte et al., 2015)¹⁵, we changed the expression “fear extinction” to “persistent attenuation of fear”, as the previous study originally described¹⁵ (**Page 2, Line 43-44**).

Based on the reviewer’s suggestion, we performed **new experiments** to examine whether the dMTm→CeA pathway is involved in fear extinction. We injected CamKIIa-NpHR3.0 virus into the dMTm and implanted optical fiber into the CeA. Then, we inhibited the dMTm-CeA pathway during fear extinction learning. The result showed that the inhibition of dMTm-CeA pathway during fear extinction learning promotes fear extinction (Response Fig. 3). This result is comparable with the previous study (Do-Monte et al., 2015)¹⁵ which showed that the optogenetic silencing of the PVT→CeA pathway at 7d after the conditioning induces persistent attenuation of fear. We added this data in the manuscript (**Fig. 5, Page 9-10, Line 203-218**).

As the reviewer pointed out, Ramanathan and colleagues (2018; Nature Communications)¹⁶ showed the importance of thalamic nucleus reuniens (RE) in fear extinction. Considering the IL projects to both the TRNrv and the RE, it seems that the IL simultaneously signals two different parts of the midline thalamus to promote fear extinction: 1) the ventral part, i.e., the RE, which would encode the information of safe context and prevent overgeneralization of conditioned fear^{16,17} and 2) the dorsal part, i.e., the dMTm, of which suppression via the TRNrv would interfere the maintenance of fear memory^{15,18}. As the reviewer suggested, we cited this work in the introduction (Page 2, Line 44), and also we discussed this issue in the discussion (Page 12-13, Line 279-285).

Response Fig. 3. The suppression of dMTm→CeA pathway during extinction learning promotes fear extinction.

Major issue 2:

TRN nuclei are rich in GABAergic neurons. However, it is unclear what proportion of cells are PV+. In Fig 3 they mention that 24-32% of cells are light-responsive, which challenges the conclusion that roughly 1/3rd of TRN cells are PV+ cells. More importantly, do all of the projections from TRN to dMT originate from PV+ cells as opposed to other GABAergic cells? The authors should either 1) answer this question quantitatively or at least cite relevant articles that support their claim that dMT projectors are PV+ cells or 2) temper the language that these effects are mediated by PV+ neurons in TRN.

The reviewer pointed out the discrepancy between the previously-known rich proportion of PV+ cells in the TRN and the low proportion (24-32%) of a photo-tagged parvalbumin (PP) neuron in our study. As the reviewer mentioned, and also as we described (Page 3, Line 72-73), the majority of TRN neurons are known to be PV-positive (98% in mice and around 70% in primates)^{19,20}. One thing we want to clarify is that the proportion of PP neuron does not reflect the proportion of PV+ cells in the TRN. Among PV+ cells in the TRN, only the neurons which satisfy the following conditions would be identified as PP neuron: 1) successfully expresses cre-dependent ChR2 virus and 2) closely located to the optical fiber to be illuminated by the light. For this reason, the proportion of PP neurons would be much less than the proportion of total PV+ neurons in the TRN.

Also, the reviewer pointed out whether our optogenetic manipulation of TRN-dMT projection was specific to PV+ cells. In our behavior (Fig. 2 and 3) and recording experiments (Fig. 3 and 4), we used PV-cre mice and cre-dependent viruses. Thus, only the PV+ cells would express the virus and the optogenetic manipulations were specific to PV+ cells.

Major issue 3:

The behavioral controls for the optogenetic experiments are unclear. The authors should describe the procedures used in the control (CTR) group, which is currently not described in the relevant figures, legends, or methods. At minimum, the authors should include an inactive virus group that receives light stimulation to control for light-induced behavioral deficits in TRNrv.

We thank the reviewer for the comment. The reviewer pointed out that the description for behavioral control is not clear. We added more detailed description in the method (**Page 18, Line 404-412**). In short, we injected identical active virus into both the control group and the experimental group and differentiated the groups by controlling the light delivery (Response Fig. 4a). As the reviewer pointed out, there is a concern that the light itself would act as a visual cue to mice and possibly affect the behavior. Also, the previous study reported that light stimulation itself can elevate the temperature leading the increase of firing activity, which is known as heating effect²¹. In our experimental setup, we carefully avoided these light-induced effects in two ways: 1) We completely blocked the light leakage by using opaque dental cement, covered-sleeve and the tube-curtain (Response Fig. 4a) as described in the method (**Page 18, Line 407-409**). 2) For the heating effect, we carefully chose the intensity of the light to be safe from the heating effect based on the previous study²¹, as we described in the method (**Page 18, Line 409-412**). Also, we confirmed that our light stimulation protocol (1.1 mW with 100 micron core fiber, or 140 mW/mm²) does not cause heating effect by recording single units with various light powers (Response Fig. 4b-d). We added this data to **Supplementary Fig. 3**.

To clearly address the reviewer's concern, we performed **new experiments** to determine whether our control group is comparable with the control group using inactive virus (Response Fig. 3, CTR1-Unstim. group and CTR2-EYFP group). As the result shows, we confirmed that the freezing levels between our control group and the inactive virus group during the extinction learning and retrieval test were comparable (Response Fig. 3). We added this data to **Fig. 5**.

Response Fig. 4. Experimental setup for optogenetic stimulation. The light power of our optogenetic experiment (1.1 mW) does not induce heating effect.

Major issue 4:

It is interesting that authors observed deficits with 10 Hz, but not 1 Hz, stimulation. The authors should clarify the rationale for using two different optical stimulation procedures and explain why they observed with one stimulation but not the other.

At the beginning of this project, because very limited information about the TRNrv was available, we needed to explore to find an effective stimulation protocol by directly testing various frequencies during the behavior. Thus, as an exploration, we tested the 1 Hz and the 10 Hz stimulation. The possible explanation for the ineffectiveness of the 1Hz could be that 1Hz is much lower than the baseline firing rate of the TRNrv (8.834 ± 1.456 Hz, Mean \pm SEM, measured during 30secs before the beginning of neutral tone test). We revised the manuscript on this issue (**Page 6, Line 127**).

Major issue 5:

In Fig 3F&G, the authors show changes in neuronal firing rates among TRN neurons in response to an auditory CS after fear conditioning. However, it is unclear how these changes relate to the hypothesized role for TRN in fear extinction. The work is not well connected to the other experiments.

The data in Fig. 3 demonstrates that the TRNrv is involved in fear extinction by increasing the firing activity, which is selectively observed in response to conditioned tone (during extinction learning) but not in response to the neutral tone. The reason why we recorded the firing rates of the TRNrv is that we observed the manipulation of firing activity of the TRNrv by optogenetic tools affect the fear extinction, leading us to hypothesize that the change of firing activity of the TRNrv would be critical for physiological fear extinction. This encouraged us to examine the firing activity of the TRNrv during extinction learning. We revised the manuscript for a better flow (**Page 6, Line 136-138**).

Major issue 6:

The authors report anatomical data showing that TRNrv neurons disynaptically project to the central amygdala (CeA) via the dMT. On this basis they suggest that the TRNrv influences extinction via this projection, however there are no functional data to support this conclusion. Importantly, the dMT itself has not been implicated in fear extinction in previous work (see #1). Moreover, dMT projections to CeA have been shown to be involved in fear conditioning and retrieval (see Padilla-Coreano et al., 2012; Do-Monte et al., 2015; Penzo et al., 2015). Hence, the authors should either provide functional evidence to support the role for TRN->dMT->CeA projections in fear extinction or remove this part of the story.

We thank the reviewer for the comment. To address the reviewer's comment, we performed new experiments as we described in the response (Major issue #1) as above. In this experiment, we injected CamKIIa-NpHR3.0 virus into the dMTm and implanted the optical fibers into the CeA, and we inhibited this dMTm-CeA pathway during extinction learning (Response Fig. 3). We observed that the inhibited group showed significantly reduced freezing level in retrieval test compared to the control groups. This result indicates that dMTm→CeA pathway is involved in fear extinction. We added this data in the manuscript (**Fig. 5, Page 9-10, Line 203-218**).

Major issue 7:

The discussion is abrupt. The authors should edit this section to improve its flow, and most importantly, should discuss the implications for a role of TRN in fear extinction in the context of other work showing that the ventral midline nuclei (including the nucleus reuniens) are important for fear extinction.

We thank the reviewer for the comment. As the reviewer suggested, we intensively revised the discussion part and added the discussion related to the thalamic nucleus reuniens (**Page 12-13, Line 279-285**).

Minor issues:

1) In the second paragraph of the methods section, the authors mention an active virus (with hCHR2) in a tracing study. Is this a typo?

It is not a typo. We used active virus because it shows strong axonal fluorescence and, as far as we know, it does not have side-effect for the tracing study compared to inactive virus.

2) In Supplemental Fig 2 the optic fiber tracts are not visible. The authors should provide a figure illustrating the localization of the optic fibers in TRN.

For the localization of the optical fiber in the TRN, it is shown in Fig. 2b. We agree that the optical fiber track is hard to be seen in the figure. We have removed this supplementary figure because the main purpose of this supplementary figure was to show that our optogenetic stimulation reliably activates the adjacent TRNrv neurons, but this was already shown by in-vivo recording (Fig. 3b).

3) The authors should report the software used for statistical analyses and the relevant p-values in the results section (not just in the figure legends).

We used commercially available software (SigmaPlot 12.0, Systat Software, Inc.) as we described in methods (**Page 22, Line 487-488**). As the reviewer suggested, we now show statistical values throughout the results section.

4) The article needs extensive proof-read to correct the extensive number of typos and grammatical and spelling errors.

As the reviewer suggested, we intensively revised typos and grammatical errors throughout the manuscript.

Reviewer #3 (Remarks to the Author):

In this paper by Lee et al., a series of experiments was conducted to investigate the role of the reticular thalamic nucleus (TRN) in extinction of conditioned fear. Anatomical tracing studies were conducted to show that rostroventral TR (TRNrv) sends inhibitory projections to the dorsomedial thalamus (dMT), which in turn projects to amygdala. Exciting TRNrv enhances fear extinction, whereas inhibiting TRNrv neurons (or inhibiting TRNrv terminals in dMT) impairs fear extinction. TRNrv neurons respond to a tone CS after it has been paired with shock, but not before. The authors argue that this pattern of results indicates a prominent role for TRNrv in the extinction of conditioned fear.

These findings are timely and would be of wide interest to the neuroscience community, especially for researchers who study the neural basis for fear and anxiety. The study impressively integrates a range of modern tools for genetic targeting of neural pathways, optogenetic stimulation, and single-unit recording. However, there are a few major issues that need be addressed (see below). First and foremost, the freezing analyses do not include a comparison of tone-evoked responses against freezing in the absence of a tone. Another issue is that some of the reported effects appear to possibly arise from differences among control groups, rather than experimental groups.

GENERAL COMMENTS

Major issue 1:

A major issue is that throughout the paper, the extinction graphs show only freezing during the tone. There is no comparison against the pre-tone baseline. It is therefore not possible to tell whether the authors are measuring conditioned freezing evoked by the tone, or persistent mobility/immobility that is lasting throughout the entire extinction session (regardless of whether the tone is on or off). It is imperative that the authors compare freezing in the presence versus absence of the tone, to validate the claim that this is conditioned freezing evoked by the tone. A plot of freezing during the pre-CS period (or of the difference score between CS and pre-CS freezing) should be shown, along with

appropriate statistics. Pre-CS freezing should be scored blindly in the same manner as freezing during the CS.

We thank the reviewer for this comment. We agree that it is critical to determine whether the freezing was evoked by the tone or by other factors like persistent mobility/immobility. In our experimental setup, the conditioning was performed in context A (square box with shock grid in conditioning chamber) which is totally different from the context B (white cylindrical acrylic box in extinction chamber) in which the extinction learning was performed. In this setup, we do not observe any significant baseline freezing in extinction learning, thus we did not record baseline freezing, as we did not in our previous publications^{22,23}. Nonetheless, to address the reviewer's concern, we performed **new experiments** in which we recorded the baseline freezing levels before the extinction learning and before the retrieval test (Response Fig. 1). In this experiment, for close comparison, we used the mice of same background (B6/129 F1) as used in the previous optogenetic experiments (Fig. 2 and 4). As shown in result, we did not observe significant increase of the baseline freezing levels before the extinction and the retrieval test compared to the freezing levels in the first tone of conditioning (two-way RM ANOVA, $F_{(2,34)} = 1.300$, $P = 0.286$). For this reason, we believe that the freezing response shown in Fig. 2 is a reliable measure of conditioned freezing evoked by the tone. We hope the reviewer agrees with our opinion. We added this data to **Supplementary Fig. 2**.

Major issue 2:

It is also somewhat troubling that the “accelerated” extinction in the ChR2 group (Fig. 2d, Supp Fig. 3b) appears to result not only from “acceleration” of extinction in the ChR2 group, but also to poor extinction in the control group. Moreover, the “impaired” extinction in the Arch group (Fig. 2f) is obtained by comparison against a control group which exhibits lower freezing levels than other control groups (especially those from the ChR2 experiments). It looks like the Arch group in Fig. 2f would not show impaired extinction if it were compared against the control group for ChR2, and it is not clear that the ChR2 group in Fig. 2d would show significantly enhanced extinction if it were compared against the control group for Arch in Fig. 2f. It is a bit of a red flag for me when the findings of a study appear to be attributable to variations in the control groups, rather than the experimental groups. The findings would be more convincing if the data from control groups were more consistent across experiments.

The reviewer pointed out that the freezing levels of the control groups are different. In the experiment of optogenetic excitation (Fig. 2d), we delivered a stronger shock (0.5mA) compared to the 0.3mA shock used in the inhibition (Fig. 2f and 2h) as designated in the figure (at the bottom of the Day1 in each Fig. 2d, 2f and 2h). The reason we applied the stronger shock in the excitation experiment was to avoid floor effect, allowing clear detection of freezing reduction, as previous study employed¹. That is, because we observed that 0.3mA shock induces mild freezing level during extinction learning and retrieval test, we increased intensity of shock to see clear reduction in freezing levels. For similar reason, we applied a weaker shock (0.3mA) in inhibition experiment (Fig. 2f and 2h) to avoid ceiling effect¹. We now describe this in the main text (**Page 6, Line 123-127**). In addition, to clearly show how the intensity of shock affects freezing levels in our setup, we performed **new experiments** with two different shock intensities (0.3mA and 0.7mA) (Response Fig. 1). We used the mice of same background (B6/129 F1) as used in the previous optogenetic experiments (Fig. 2). As expected, the mice with a stronger shock showed significantly increased freezing levels throughout the extinction learning on Day2 and the retrieval test on Day3 (Response Fig. 1). We added this data to **Supplementary Fig. 2** and revised the manuscript (**Page 6, Line 124-125**).

Major issue 3:

One large piece that seems to be missing from this story is what happens when TRN is manipulated during fear acquisition, rather than extinction. Would stimulation of TRNr_v impair the acquisition of conditioned fear? The title of the paper implies that TRNr_v plays some specialized or specific role in fear extinction. But we don't really know this unless acquisition experiments are performed for comparison.

We agree with the reviewer. It is unknown whether the TRNr_v is involved in fear acquisition. To address the reviewer's comment, we performed new experiments. We injected DIO-ChR2 virus and implanted optical fibers into the TRNr_v and stimulated the TRNr_v during the tone presentations of the fear conditioning. The result showed that the excitation of the TRNr_v during the fear conditioning does not affect the fear acquisition (Response Fig. 5). Considering that the TRNr_v provides inhibitory inputs to the dMTm, our result is consistent with the previous report that the chemical inhibition of the dMT during fear conditioning does not affect fear acquisition¹⁴. We added this data to Supplementary Fig. 4 and revised the manuscript (Page 5, Line 116-119).

Response Fig. 5. The optogenetic excitation of the TRNr_v during fear conditioning does not affect fear acquisition.

SPECIFIC COMMENTS

The disynaptic tracing data in Fig.5 is quite beautiful, but how many mice has this pattern been observed in? It appears to be n=1 but this is not entirely clear. Replication in additional mice might be advisable. The sample size for the tracing data in Fig. 1 also seems rather small, but in that case, the problem is ameliorated by showing the reversed labeling for the retrograde tracers in the supplementary figures. 2-3 replications of the disynaptic labeling results would be advisable.

As the reviewer suggested, we performed additional tracing experiment and our previous result was successfully replicated in two additional mice (Response Fig. 6). We added this data to the manuscript (Supplementary Fig. 8, Page 10, Line 228-229).

Response Fig. 6. TRNrV neurons are disinaptically connected to the CeA through the dMTm. Replications for the data in Fig. 6.

Are the data in Fig 2d and supplementary Figure 3b completely independent replications of the extinction experiment (no overlapping mice between these experiments)?

Yes. Those experiments are completely independent. The supplemental data was acquired in a different facility, which is located in a different city, with the mice that were raised in that facility.

Supplementary figure 2 is intended to show positioning of optical fibers in TRNrV and TRNrD, and optogenetically induced c-fos expression. Blue dashed lines are used to show the outline of the fiber position, but there does not to be any fiber track in the outlined tissue. How was it determined that the fiber was indeed positioned in this location?

We determined the fiber position based on the trajectory of the fiber in the adjacent slices. We agree that the optical fiber track is hard to be seen in the figure. We have removed this supplementary figure because the main purpose of this supplementary figure was to show that our optogenetic stimulation reliably activates the adjacent TRNrv neurons, but this was already shown by in-vivo recording (Fig. 3b). For the localization of the optical fiber in the TRN, it is shown in Fig. 2b.

How many mice were the cell recordings shown in Fig. 4 obtained from? The cells counts are rather low. A histology figure showing the electrode tracks would help to support the claim that these cells were all recorded from DMTm.

Originally we recorded from two mice. To address the reviewer's concern, we recorded three more mice and increased the number of units (total 5 mice, 32 units). We have shown the examples of newly recorded dMTm units in Response Fig. 7. As the number increased, the proportion of inhibited/not changed units were slightly changed from 72.2 / 27.8 to 78.1 / 21.9, and we revised the manuscript (Page 9, Line 190-192). As the reviewer suggested, we added the histology figure to Fig. 4a showing the position of the tetrode.

Response Fig. 7. The examples of newly recorded dMTm neurons which were suppressed by the optogenetic excitation of the TRNrv.

The first 2-3 paragraphs of the introduction could do a better job of introducing background on the TRN, for example, that it contains GABAergic projection neurons that inhibit other thalamic nuclei.

As the reviewer suggested, we added the background related to the TRN in the introduction (Page 2, Line 44-47).

p. 3: "Guided by this connection map we carried out circuit analysis and found that TRNrv neurons are activated during extinction learning, which leads to the suppression of the dMTm neurons, resulting in fear extinction." The last bit is an interpretation, not a finding, no?

The reviewer correctly pointed out that we could not directly state that the suppression of dMTm neurons results in fear extinction, with the information we had in the first submission. Now we provide the new data which shows the inhibition of dMTm→CeA pathway during extinction learning promotes fear extinction (Response Fig. 3). We added this data to Fig. 5, and we revised the manuscript based on the new result and the reviewer's comment (Page 3, Line 61-65).

p. 3: “As a result, we observed axonal signals across the dMT (Fig. 1d).” This is poorly worded and confusing. Better to say “After virus injections into TRN, we observed GFP labeled axons throughout the dMT (Fig. 1d).”

As the reviewer suggested, we revised the expression (**Page 4, Line 77-78**).

p. 4: “and with precise positioning of optical fiber (Fig. 1c, 1e and 1g).” It appears that this should say Fig. 2c, 2e, and 2g?

We apologize for the mistake. we revised the manuscript (**Page 4, Line 96**).

p. 6: “As a result, surprisingly, TRNrv neurons showed increased firing activity selectively to the conditioned tones (Fig. 3f, first row, ALL neurons) but not to the neutral tones (Fig. 3d, first row, ALL neurons).” The word “surprisingly” seems inappropriate here. Isn't this what the authors were expecting?

The reviewer is correct. That result was what we expected. We revised the expression “surprisingly” to “notably” (**Page 7, Line 152**).

p. 8: “we examined that whether” should say “we examined whether”

As the reviewer pointed out, we changed the expression (**Page 10, Line 221-222**).

In some places, the authors write “excitated” where they should write “excited” (“excitated” is not a word).

As the reviewer pointed out, we revised the manuscript (**Page 9, Line 190 and 193**).

The summary table for statistics in the supplement only gives omnibus F values for the ANOVAs. It would be nice to see the main and interaction effects as well.

As the reviewer suggested, we added the information for main and interaction effects (**Supplementary Table 1**).

p. 9: “Considering the well-known factor that the sensory sector of the TRN intimately communicates not only with sensory thalamus but also with corresponding sensory cortex 16, which cortical area could correspond to TRNrv-dMTm circuit?” This question is poorly worded and hard to understand. I assume that the authors are proposing future research to investigate which cortical regions are reciprocally connected to the TRNrv-dMTm circuit, since each thalamic region tends to be mated with particular cortical areas. But the wording of this entire paragraph is confusing and awkward. Later in the paragraph, it is stated “it may be reasonable to select the infralimbic cortex as upstream brain area of the TRNrv,” as if it is up to the authors to choose which cortical regions are connected to TRNrv-dMTm. This paragraph should be rewritten.

We thank the reviewer for the comments. As the reviewer suggested, we rewrite the discussion to clearly deliver the message (**Page 12, Line 265-275**).

The discussion should address question about what functional role the TRNrv-dMTm circuit might play in extinction. Does this circuit simply inhibit fear and thereby contribute to a sense of safety? Is it involved in generating negative prediction errors that might drive extinction learning? Are there experiments that might be done to dissociate between these and other possibilities?

As the reviewer suggested, we added the discussion for this issue (**Page 11, Line 246-251**). The previous study has shown that the PVT, which is a part of the dMTm, does not encode prediction error in aversive learning²⁴. Thus, the facilitation of the extinction by TRNrv→dMTm pathway might not be relevant to a generation of negative prediction errors for the conditioned tones. Rather, it could be because the TRNrv dampens the salience of the aversive tone encoded by the PVT²⁴, which might allow safety information to be encoded better in other extinction-related circuits – for example, the infralimbic cortex→the thalamic nucleus reuniens pathway¹⁶.

References

1. Adhikari, A. *et al.* Basomedial amygdala mediates top-down control of anxiety and fear. *Nature* **527**, 179–185 (2015).
2. Máttyás, F., Lee, J., Shin, H.-S. & Acsády, L. The fear circuit of the mouse forebrain: connections between the mediodorsal thalamus, frontal cortices and basolateral amygdala. *Eur. J. Neurosci.* n/a-n/a (2013). doi:10.1111/ejn.12610
3. Mahn, M., Prigge, M., Ron, S., Levy, R. & Yizhar, O. Biophysical constraints of optogenetic inhibition at presynaptic terminals. *Nat. Neurosci.* **19**, 554–556 (2016).
4. Cornwall, J., Cooper, J. & Phillipson, O. Projections to the rostral reticular thalamic nucleus in the rat. *Exp. Brain Res.* **80**, 157–171 (1990).
5. Do-Monte, F. H., Manzano-Nieves, G., Quiñones-Laracuente, K., Ramos-Medina, L. & Quirk, G. J. Revisiting the Role of Infralimbic Cortex in Fear Extinction with Optogenetics. *J. Neurosci.* **35**, 3607–3615 (2015).
6. Milad, M. R. & Quirk, G. J. Neurons in medial prefrontal cortex signal memory for fear extinction. *Nature* (2002).
7. Vidal-Gonzalez, I., Vidal-Gonzalez, B., Rauch, S. L. & Quirk, G. J. Microstimulation reveals opposing influences of prelimbic and infralimbic cortex on the expression of conditioned fear. *Learn. Mem.* **13**, 728–733 (2006).
8. Peters, J., Dieppa-Perea, L. M., Melendez, L. M. & Quirk, G. J. Induction of Fear Extinction with

- Hippocampal-Infralimbic BDNF. *Science* **328**, 1288–1290 (2010).
9. Sierra-Mercado, D., Padilla-Coreano, N. & Quirk, G. J. Dissociable roles of prelimbic and infralimbic cortices, ventral hippocampus, and basolateral amygdala in the expression and extinction of conditioned fear. *Neuropsychopharmacology* **36**, 529–538 (2010).
 10. Rhodes, S. E. V. & Killcross, S. Lesions of Rat Infralimbic Cortex Enhance Recovery and Reinstatement of an Appetitive Pavlovian Response. *Learn. Mem.* **11**, 611–616 (2004).
 11. West, E. A., Saddoris, M. P., Kerfoot, E. C. & Carelli, R. M. Prelimbic and infralimbic cortical regions differentially encode cocaine-associated stimuli and cocaine-seeking before and following abstinence. *Eur. J. Neurosci.* **39**, 1891–1902 (2014).
 12. Laurent, V. & Westbrook, R. F. Inactivation of the infralimbic but not the prelimbic cortex impairs consolidation and retrieval of fear extinction. *Learn. Mem.* **16**, 520–529 (2009).
 13. Thompson, B. M. *et al.* Activation of the infralimbic cortex in a fear context enhances extinction learning. *Learn. Mem.* **17**, 591–599 (2010).
 14. Padilla-Coreano, N., Do-Monte, F. H. & Quirk, G. J. A time-dependent role of midline thalamic nuclei in the retrieval of fear memory. *Neuropharmacology* **62**, 457–463 (2012).
 15. Do-Monte, F. H., Quiñones-Laracuente, K. & Quirk, G. J. A temporal shift in the circuits mediating retrieval of fear memory. *Nature* **519**, 460–463 (2015).
 16. Ramanathan, K. R., Jin, J., Giustino, T. F., Payne, M. R. & Maren, S. Prefrontal projections to the thalamic nucleus reuniens mediate fear extinction. *Nat. Commun.* **9**, 4527 (2018).
 17. Xu, W. & Südhof, T. C. A Neural Circuit for Memory Specificity and Generalization. *Science* **339**, 1290–1295 (2013).
 18. Penzo, M. A. *et al.* The paraventricular thalamus controls a central amygdala fear circuit. *Nature* (2015). doi:10.1038/nature13978
 19. Zikopoulos, B. Prefrontal Projections to the Thalamic Reticular Nucleus form a Unique Circuit for Attentional Mechanisms. *J. Neurosci.* **26**, 7348–7361 (2006).
 20. Hou, G., Smith, A. G. & Zhang, Z.-W. Lack of Intrinsic GABAergic Connections in the Thalamic Reticular Nucleus of the Mouse. *J. Neurosci.* **36**, 7246–7252 (2016).

21. Stujenske, J. M., Spellman, T. & Gordon, J. A. Modeling the Spatiotemporal Dynamics of Light and Heat Propagation for In Vivo Optogenetics. *Cell Rep.* **12**, 525–534 (2015).
22. Lee, S. *et al.* Bidirectional modulation of fear extinction by mediodorsal thalamic firing in mice. *Nat. Neurosci.* (2011). doi:10.1038/nn.2999
23. Baek, J. *et al.* Neural circuits underlying a psychotherapeutic regimen for fear disorders. *Nature* **566**, 339 (2019).
24. Zhu, Y. *et al.* Dynamic salience processing in paraventricular thalamus gates associative learning. *Science* **362**, 423–429 (2018).

Reviewers' Comments:

Reviewer #1:

Remarks to the Author:

The authors addressed satisfactorily all issues raised by the reviewers.

Reviewer #2:

Remarks to the Author:

The reviewers have done an outstanding job of addressing the concerns raised in the previous round of review and the manuscript is suitable for publication.

Reviewer #3:

Remarks to the Author:

In this paper by Lee et al., a series of experiments was conducted to investigate the role of the reticular thalamic nucleus (TRN) in extinction of conditioned fear. Anatomical tracing studies were conducted to show that rostroventral TR (TRNrv) sends inhibitory projections to the dorsomedial thalamus (dMT), which in turn projects to amygdala. Exciting TRNrv enhances fear extinction, whereas inhibiting TRNrv neurons (or inhibiting TRNrv terminals in dMT) impairs fear extinction. TRNrv neurons respond to a tone CS after it has been paired with shock, but not before. The authors argue that this pattern of results indicates a prominent role for TRNrv in the extinction of conditioned fear.

These findings are timely and will be of wide interest to the neuroscience community, especially for researchers who study the neural basis for fear and anxiety. The study impressively integrates a range of modern tools for genetic targeting of neural pathways, optogenetic stimulation, and single-unit recording.

All of my prior concerns have been adequately addressed by thorough replies, appropriate addition of new data, and revisions to the manuscript text.